

# Separating and Quantifying Facility-Level Methane Emissions with Overlapping Plumes for Spaceborne Methane Monitoring

Yiguo Pang[1,2], Longfei Tian[1], Denghui Hu[1], Shuang Gao[1], and Guohua Liu[1,2]

[1]Innovation Academy for Microsatellites of Chinese Academy of Sciences, Shanghai China.
[2]University of Chinese Academy of Sciences, Beijing, China.

**Correspondence:** Guohua Liu (liugh@microsate.com)

**Abstract.** Quantifying facility-level methane emission rates using satellites with fine spatial resolution has recently gained significant attention. However, the existing quantification algorithms usually require the methane column plume from a single point source as input. Such approaches are challenged with overlapping plumes from multiple point sources. To address these challenges, a multi-objective heuristic optimization algorithm is introduced to perform parameter estimations for the 2D multi-source Gaussian plume model, which serves as the basis for the separation method. In addition, to improve the separation performance on relatively weaker sources, we proposed a metric called local binary pattern metric (LBPM), which is only sensitive to the sign of the gradient as a minimization objective. To verify the proposed separation method, observation system simulation experiments (OSSE) of various scenarios are performed, where the integrated mass enhancement (IME) is selected as a representative single-source quantization method. The result shows that plume overlapping will increase the quantifying error of IME as overlapping pixels may not be attributed correctly; compared to unseparated overlapping plumes, the proposed separation method decreases the quantification MAPE from 1.46 to 0.45 on synthetic observation over real targets. Our separation method can separate observation of overlapping plumes from multiple sources into several observations each with a plume from a single source, thereby extending single point source quantifying algorithms, such as IME, to be applicable within scenarios of multiple point sources.

## 1 Introduction

Since the industrial revolution, the increasing anthropogenic emissions of greenhouse gases (GHG), have emerged as the foremost contributor to global warming and climate change, obstructing global sustainable development (IPCC, 2021). To tackle this challenge, the global community has united and expressed a strong will to limit long-term warming below 1.5°C above the pre-industrial level, as stipulated in the Paris Agreement under the United Nations Framework Convention on Climate Change (UNFCCC). Comprehensive monitoring of global GHG is vital for verifying human activities' impact on climate change, observing climate change trends, formulating solutions to address climate change, and evaluating the efficacy of climate policies. The conventional way to estimate GHG emissions is to multiply the elements of human activities by emission factors, using statistical methods (Calvo Buendia et al., 2019). Yet, owing to the substantial uncertainty of emission factors and source coverage (Zhao et al., 2017; Suarez et al., 2019), the performance of this bottom-up method is limited. In this



regard spaceborne GHG monitoring capabilities, e.g., OCO-2/3 (Nassar et al., 2017), TROPOMI (Zhang et al., 2020) and TanSat (Yang et al., 2023), have demonstrated their ability to quantify anthropogenic GHG emissions from large sources, such as cities and large thermal power plants, which are considered point sources. Spaceborne GHG monitoring is capable of undertaking independent, objective, and high spatiotemporal coverage measuring, and is thus considered important to verify the accuracy of bottom-up GHG emission inventories (Calvo Buendia et al., 2019; Liu et al., 2022).

Methane ($CH_4$) is a greenhouse gas second only to carbon dioxide($CO_2$) in terms of radiative forcing, with a global warming potential (GWP-100) of about 27-29 times that of $CO_2$ on unit emission and a lifespan of only about 11.8 years (IPCC, 2021). As a result, taking proactive measures to reduce anthropogenic methane emissions can help alleviate global warming in the short term. Research shows that anthropogenic methane emissions are mostly concentrated at a few point sources with high emission rates (Nisbet et al., 2020; Cusworth et al., 2020; Duren et al., 2019; Frankenberg et al., 2016). Additionally, an
observed methane plume usually has a higher signal-to-noise ratio (SNR) given the same emission rate, as the background concentration of methane (approximately 1.8 ppm) is generally much lower compared to $CO_2$ (approximately 420 ppm). These features provide convenience for spaceborne monitoring of anthropogenic methane emissions. A recent trend is monitoring methane point sources using orbital instruments with fine spatial resolution, as smaller pixels are generally more sensitive to dry column concentration of point sources with relatively low emissions rates (Jervis et al., 2021). Remarkable processes have
been achieved by these systems, including a dedicated GHG point source monitoring constellation named GHGSat (Varon et al., 2018, 2019; Jervis et al., 2021), hyperspectral satellites such as PRISMA (Guanter et al., 2021) and EnMAP (Green et al., 2020), as well as multispectral satellites such as Sentinel-2 (Zhang et al., 2022; Gorroño et al., 2022; Ehret et al., 2022) and WorldView-3 (Sánchez-García et al., 2022).

One of the primary purposes of spaceborne methane point source monitoring is to quantify the emission rates. To do so,
a widely used method is spaceborne measuring backscattered solar radiation in visible and shortwave infrared (VSWIR), followed by the application of inversion algorithms, such as the optimal estimation theory based atmospheric inversion methods (Rodgers, 2000; Frankenberg et al., 2005; Jervis et al., 2021), matched filter (Thorpe et al., 2014) and deep learning based methods (Özdemir and Koz, 2023), to retrieve the column concentration or the column enhancement of methane. Quantification methods, including Gaussian plume fitting (Bovensmann et al., 2010; Nassar et al., 2017, 2021), integrated mass enhancement
(IME; Frankenberg et al., 2016; Varon et al., 2018) method and Cross-sectional Flux (CSF; White et al.; 2011) method, are then applied in conjunction with meteorological priors to estimate the emission rates of the methane point source. These quantification methods usually rely on the detection or masking of plumes from observations. To tackle this issue, Bovensmann et al. (2010) estimates the $XCO_2$ backgrounds using the entire observations as inputs; Nassar et al. (2017) distinct the plume and the backgrounds with a 1 % density cutoff criteria; Kuhlmann et al. (2019) proposes a $Z$ test based plume detection algorithm
to mask pixels with statistically higher values as the inputs; Varon et al. (2018) combines Student's $t$ test with computer vision (CV) based methods to detect plume pixels. Besides, deep learning methods are employed to perform quantification (Jongaramrungruang et al., 2022) or even end-to-end detection and quantification (Joyce et al., 2022).

However, few research consider quantifying emissions from overlapping plumes of multiple spatially adjacent sources. Plume overlapping poses a challenge to quantification as it breaks the one-to-one correspondence between a plume pixel and a





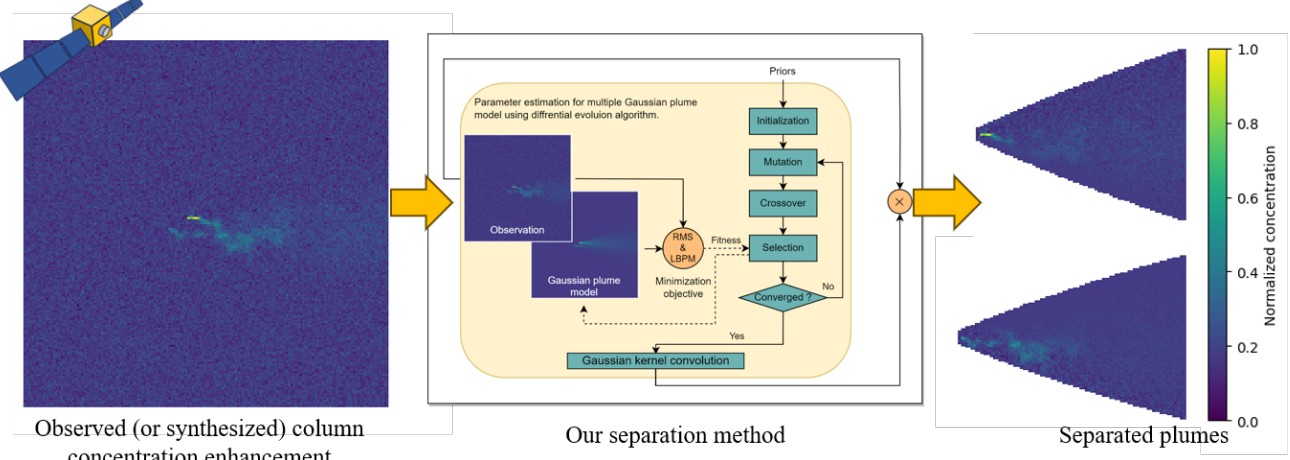

Observed (or synthesized) column concentration enhancement

Our separation method

Separated plumes

**Figure 1.** Proposed methodology for separating overlapping plumes. A heuristic optimization-based method is proposed to separate the overlapping plumes, which extends the traditional IME method to be applicable in multi-source emission scenarios. This method takes observed methane column enhancement and priors with uncertainty as inputs and outputs the separated plumes for the following quantification.

source, which means the mass in each detected plume pixel may originate from multiple sources, introducing additional errors in quantification when the mass in a conjoint pixel is attributed to any single source. In many cases, plume overlap increases the difficulty of quantification (Kuhlmann et al., 2019), or even makes quantification impossible (Duren et al., 2019; Kuhlmann et al., 2020; Sánchez-García et al., 2022). Given that the median interval distances of potential methane sources in California, US, as we analyzed using VISTA-CA inventory (Hopkins et al., 2019) are less than 200 m, respectively, plume overlap can be

not uncommon. This statement can be especially valid as the plume length for a typical 30m fine resolution methane monitoring satellite is about 67-2200 m (Ayasse et al., 2019). Therefore, it is necessary to separate the overlapping plumes and to ensure one-to-one matches between plume pixels and sources, in order to extend the single point source quantification algorithms to multiple sources scenarios. Nassar et al. (2021, 2017) use the Gaussian plume model to model the interference sources as background and then applies the Gaussian plume fitting to estimate the emission rate of the primary source. However, this

method requires priors from inventories for interference sources, which are mostly difficult to access for facility-level methane quantification.

In the past, there has been extensive research on quantifying the emission rates of atmospheric pollutant sources using ground-based sensor networks (Hutchinson et al., 2017). Due to the similarities of the physical mechanism between pollutants and GHG dispersion in the atmospheric boundary layer (ABL), source term estimation methods for atmospheric pollutants dispersion have been introduced into spaceborne GHG source monitoring, e.g., the CSF method (White et al., 1976; Krings

et al., 2011). Multiple source estimation has also been studied in this domain, e.g., Lushi and Stockie (2010) uses a Gaussian plume model to estimate the emission rates of multiple point source pollutants by fitting sensor measurements with linear least squares. however, the high nonlinearity of the Gaussian plume model when priors, such as wind speed, wind direction,





and source locations, are uncertain, may result in estimation failure. This limitation can severely impact the performance of

spaceborne measurements, where obtaining accurate priors may be difficult. One solution is to use heuristics optimization algorithms, such as genetic algorithms, to fit a Gaussian plume model to sensor measurements and estimate the emission rate (Allen et al., 2007; Haupt et al., 2007). This method can simultaneously estimate these priors, making the estimation results robust to priors.

To quantify facility-level methane emissions with overlapping plumes, as shown in Fig. 1, a heuristic optimization algo-

rithm is introduced to estimate the parameters of the 2-dimensional (2D) Gaussian plume model for simulated observations and then separate overlapping plumes into a set of plumes from single sources, thus expanding the applicability of the single source quantification method (e.g., the IME method) to be applicable for quantifying emission rates of overlapping plumes from multiple point sources. In addition, a local binary pattern metric (LBPM) is proposed as one of the minimization objectives for the optimization algorithm. The LBPM, only sensitive to the gradient sign of the observed concentration, along

with the conventional root mean square (RMS) metric, is used for the multi-objective optimization to improve the estimation performance on sources with relatively lower emission rates. To study the impact of overlap on quantification and evaluate the proposed separation method, we simulate plumes using large eddy simulations with the Weather and Research Forecasting Model (WRF-LES; https://www.mmm.ucar.edu/models/wrf). Then, observations of various sources, meteorological and observation characteristics are synthesized based on these plumes. Three sets of observation system simulation experiments (OSSE)

were performed, including ideal single source, ideal dual sources and real scenarios. In these experiments, the improvements of the separation algorithm in missed detection and quantization error compared with the conventional single point source quantification algorithm are evaluated. Finally, we also test our method on authentic satellite observed overlapping plumes.

## 2 Methods

### 2.1 Separating method for overlapping plumes of multiple sources

In this section, the dispersion of methane emissions was modelled using a 2D multi-point Gaussian plume model (Section 2.1.1). A multi-objective heuristic optimization algorithm is introduced to separate overlapping plumes to perform parameter estimations for the dispersion model by minimizing the metrics between the modelled and observed column mass images (Section 2.1.2). Furthermore, to improve the estimation performance on relatively weaker sources, a metric called LBPM is proposed for the minimization (Section 2.1.3).

### 105 2.1.1 2-D multi-source Gaussian plumes dispersion model

The transport mechanism of methane from point sources in the ABL can be very complicated, as affected by multiple factors such as atmospheric turbulence, chemical reactions and terrain effects. From the perspective of mass conservation, considering wind transport, gradient diffusion, and source-sink terms, the convection-diffusion equation can be obtained to represent this



mechanism, which can be written as (Stockie, 2011)

$$\frac{\partial C}{\partial t} + \nabla \cdot (C\boldsymbol{u}) = \nabla \cdot (\mathbf{K}\nabla C) + S, \tag{1}$$

where $C$ represents the methane column mass concentration at a certain moment; $t$ represents time; $\boldsymbol{u}$ represents the 2D field of wind velocity vectors; matrix $\mathbf{K}$ is diagonal, with its elements representing diffusion coefficients for each wind velocity direction; $S$ represents the source item.

One way to solve this partial differential equation (PDE) is the numerical method (e.g., Hosseini and Stockie, 2017), however,
the computational cost can be enormous. Analytical methods, on the other hand, simplify the problem by making assumptions, allowing for the derivation of analytical solutions to the PDE. For instance, the Gaussian plume expression of a point source can be obtained from the convection-diffusion equation by assuming that the wind speed is constant and uniform, the emission rate is time-invariant and the turbulence is negligible (Sutton, 1932; Ermak, 1977; Stockie, 2011). The Gaussian plume model is widely applied to describe the pollutants, as well as the GHG dispersion in ABL, particularly in spaceborne GHG moni-
toring research (Bovensmann et al., 2010; Nassar et al., 2021; Jacob et al., 2022). As a result, we modelled the column mass concentration(kg/m$^2$) at the location $(x, y)$ using a 2D Gaussian plume model for a ground-level point source, which can be written as (Sutton, 1932; Bovensmann et al., 2010)

$$C(x, y) = \frac{Q}{\sqrt{2\pi}\sigma_y(x)u} \exp\left(-\frac{1}{2}\left(\frac{y}{\sigma_y(x)}\right)^2\right), \tag{2}$$

where the x-axis is aligned with the direction of wind speed; $Q$ represents the emission rate (kg s$^{-1}$); $\sigma_x$ and $\sigma_y$, which are
functions (Briggs, 1973; Griffiths, 1994) of Pasquill stability class (Pasquill, 1961), represent the diffusion coefficients downwind and across-wind, respectively; $u$ represents the horizontal wind velocity (m s$^{-1}$). Eq.2 then can be extended to multiple source scenarios of $N$ sources and is given by

$$C_N(x, y) = \sum_{n=1}^{N} C_n(x', y'), \tag{3}$$

where

$$\begin{bmatrix} x' \\ y' \end{bmatrix} = \begin{bmatrix} \cos\theta & \sin\theta \\ -\sin\theta & \cos\theta \end{bmatrix} \begin{bmatrix} x - x_n \\ y - y_n \end{bmatrix}. \tag{4}$$

Here $C_n$ denotes the concentration increment at $(x, y)$ caused by the emission of point source $n$ at coordinates $(x_n, y_n)$; $\theta$ represents the wind speed angle to the x-axis in an easting/northing Cartesian coordinate system, where 0° represents west wind, 90° represents south wind, etc.

### 2.1.2 Overlapping plume separation method based on multi-objective heuristic optimization

Heuristics optimization algorithms are capable of global searching in optimization and are thus widely used for solving optimization problems. Heuristics optimization algorithms have been widely used in parameter estimation of point source dispersion models (Hutchinson et al., 2017), e.g., Allen et al. (2007), Haupt et al. (2007) and Cervone et al. (2010), showing



more robust performance compared to other optimization methods such as Bayesian inference (Platt and DeRiggi, 2012). The differential evolution algorithm (Storn and Price, 1997) is a heuristic optimization algorithm inspired by the evolution theory of biological species. Due to its ability to perform global optimization for systems with multiple continuous parameters, the differential evolution algorithm is thus an ideal candidate for parameter estimation of the dispersion model discussed in Section 2.1.1.

In this study, the differential evolution algorithm is selected as the estimation algorithm to iteratively minimize the metrics between the modelled concentration image by Eq.3 and the observed concentration image, to estimate the parameters of the dispersion model.

Here, the estimating parameters consist of source locations $(x_i, y_i)$ and emission rates $Q_i$ of source $i$, the global wind angle $\theta$ and wind velocity $u$. For the application of the differential evolution algorithm in this paper, the searching spaces for the estimating parameters are set as follows: prior ground truths $\pm$ 100 m for source locations $(x_i, y_i)$; prior ground truth $\pm$ 50% for the wind velocity, higher than the average errors of the common used reanalysis meteorological database analyzed by Varon et al. (2018) and Duren et al. (2019); prior ground truth $\pm$ 45° for the wind angle $\theta$; $0 - 5000$ kg h$^{-1}$ for emission rates $Q_i$, covering all methane point sources in Duren et al. (2019).

The minimization objective is another important part to apply the optimization algorithm. The most widely used minimization objective is to minimize the root mean square (RMS) metrics between modelled and observed concentration images. The RMS metric is given as

$$\mathcal{L}_{\mathrm{RMS}} = \sqrt{\sum_{i,j} \left( I_{\mathrm{model}}(i,j) - I_{\mathrm{obs}}(i,j) \right)^2}, \qquad (5)$$

where $I_{\mathrm{model}}(i,j)$ and $I_{\mathrm{obs}}(i,j)$ represent the modelled and observed concentration images, respectively; $i$ and $j$ represent the pixel indexes in row and column, respectively. In addition to minimizing RMS, a shape-sensitive metric called local binary pattern metrics (LBPM, see Section 2.1.3), which is sensitive to the gradient sign, is proposed to compensate for the suboptimal performance of the RMS metric in capturing the weaker source of unbalanced multi-source scenarios during the early stage of iterations. To combine the advantages of LBPM in fitting shapes during the early iteration stages and RMS in fitting absolute values during the later iteration stages, a weighted metric is proposed for the multi-objective optimization, given by

$$\mathcal{L} = \mathcal{L}_{\mathrm{RMS}} + e^{\alpha \cdot t} \cdot \mathcal{L}_{\mathrm{LBPM}}, \qquad (6)$$

where the exponential decay weight $e^{\alpha \cdot t}$ is used to leverage two metrics; $\alpha$ is the decay rate constant, which is set as 1.5; $t$ represents the number of iterations.

In the application of the differential evolution algorithm, the mutation strategy is set as best-guided mutation, i.e., DE2 in (Storn and Price, 1997); the number of population $(NP)$ is set as $10 \times (N \times 3 + 2)$, where $N$ represents the number of sources; 3 and 2 represent the numbers of parameters to be estimated for each source and entire observation, respectively; The mutation constant $F$ is set as 1 and the cross-over constant $CR$ is set as 0.9 according to Storn and Price (1997); the relative convergence criteria is set as $10^{-3}$.





The parameter estimations of the dispersion model (Eq.3) are obtained upon the convergence of the differential algorithm. To separate the overlapping plumes, the dispersion model is firstly run with the estimated parameter, producing a set of Gaussian plume concentration images for each source. Next, a 2D Gaussian kernel is applied to convolve these images in order to increase robustness against the deviations of observations, LES simulated transient plumes, from the stationary Gaussian plume. Finally, the convolved plume image for each source is normalized to the summation of plume images by each pixel, resulting in a

probability distribution function (PDF) to separate the overlapping plumes on a per-pixel basis. The separated plume image $\hat{I}_{\text{obs},n}$ of source $n$ from observation $I_{\text{obs}}$ is given by

$$\hat{I}_{\text{obs},n} = I_{\text{obs}} \cdot \frac{\hat{C}_n}{\sum_{p=1}^{N} \hat{C}_p}, \tag{7}$$

where $\hat{C}_n, \hat{C}_p$ represents the modelled and convolved plume image of source $n$ and $p$, respectively.

### 2.1.3 LBPM - a shape sensitive metric for 2D images

Haupt et al. (2007) choose RMS as the minimization metric for source parameter estimation after testing several $L_p$-norm metrics. However, the $L_p$-norm metrics, including RMS (i.e. $L_2$-norm) metrics, are more sensitive to data with higher values. This results in a preference towards fitting pixels with higher concentration, which means that sources with higher emission rates are prioritized over weaker sources with low emission rates, resulting in unequal fitting. A metric only sensitive to pixel variation trends instead of pixel absolute values is thus required to tackle this issue.

Inspired by the local binary patterns (LBP) descriptor (Ojala et al., 2002) in the CV community, we propose the local binary patterns metric (LBPM) to evaluate the difference between modelled and observed concentration images. An LBP descriptor is a binary sequence, generated by the relative relationships between the values of a central pixel and its neighbouring pixels, to represent the local shape. The LBP descriptors are only sensitive to the relative brightness relationship between pixels, meaning they are sensitive to variation trends but robust to absolute values of pixels.

To construct the LBPM, firstly, LBP descriptors need to be generated for each pixel. Given concentration in the central pixel as $g_c$, with $P$ neighbouring pixels $\{g_1, g_2, \cdots, g_P\}$, the LBP descriptor of the central pixel is constructed by (Ojala et al., 2002)

$$T = \{s(g_1 - g_c), s(g_2 - g_c), \cdots, s(g_P - g_c)\}, \tag{8}$$

where the constructing function $s(x)$ is originally defined as sign function $\text{sgn}(x)$, given by

$$\text{sgn}(x) = \begin{cases} 0, & x < 0 \\ 1, & x \geq 0. \end{cases} \tag{9}$$

Here, considering pixel value uncertainty $\sigma$ (i.e., the concentration uncertainty of observation), we define the constructing function $s(x)$ by $s_\sigma(x) = \text{sgn}(x - \sigma)$.

The following shifting procedure in Ojala et al. (2002) is omitted as rotation robustness is not desired in this research. LBPM is then obtained from the LBP descriptors. Let $T^p$ represent the $p$th element in an LBP sequence, $p \in [1, P]$. For 2 given LBP



sequences $T_a, T_b$ of two pixels, their LBPM is defined as their Jaccard index, given by

$$\text{LBPM}(T_a, T_b) = \frac{1}{P}\sum_{p=1}^{P}(1 - s_\sigma(|T_a^p - T_b^p|)), \tag{10}$$

where $T_a^p, T_b^p$ represent the $p$th element of LBP descriptors $T_a$ and $T_b$, respectively. Here, $P$ is set as 8, meaning an LBP sequence is generated with the 8 connective pixels from a centre pixel. The order of the neighbouring pixels is arbitrary yet consistent. Similarly, the LBPM between two images can also be defined as the Jaccard index of two combination sequences, each containing LPB descriptors for every pixel in its respective image, given by

$$\mathcal{L}_{\text{LBPM}}(I_a, I_b) = \sum_{i,j}\text{LBPM}\Big(T_a(i,j), T_b(i,j)\Big), \tag{11}$$

where $T_a(i,j)$ and $T_b(i,j)$ represent the LBP sequences for pixel $(i,j)$ in image $I_a$ and $I_b$, respectively.

## 2.2 Synthesized observation

### 2.2.1 Methane plume simulation

Although there have been abundant spaceborne methane observations, these observations suffer from the demerit of the lack of accurate priors, challenging the evaluation of proposed methods. Therefore, simulated observations are produced for evaluation. The large eddy simulation (LES), a promising methodology for solving the Navier-Stokes equation, is widely employed to simulate the dispersion in the ABL (Stoll et al., 2020), with results well agreed with observations (Rybchuk et al., 2021; Brunner et al., 2023). The LES run by WRF, a widely adopted atmospheric numerical simulation model framework, is thus a preferred option for spaceborne GHG monitoring (Varon et al., 2018; Cusworth et al., 2019; Brunner et al., 2023).

WRF-LES is used to simulate 3D volume methane concentrations (in kg m$^{-3}$) of dispersion plumes. The dispersion is simulated by solving the advection-diffusion equation (Eq.1), where the diffusion term is negligible due to the low concentration of methane in the atmosphere. The dispersion is thus can be simulated as passive tracer gas dispersion (Nottrott et al., 2014). We add a trace gas dispersion function with open boundary conditions by modifying the source code of the WRF 4.4 ideal LES experiment. Similar to Varon et al. (2018), methane plumes are simulated with a mean geostrophic wind of 1, 3, 5, 7 or 9 m s$^{-1}$; an inversion height of 500, 800 or 1100 m; a simulation region of 3.5 km x 6 km (across and along wind) with horizontal and vertical resolutions of 20 m and 10 m, respectively. The initial temperature is set as 293 K in the mixed layer, with a lapse rate of 0.12 K m$^{-1}$ above the inversion height. The surface sensible heat flux is set as 100 W m$^{-2}$, respectively. The model is run for 3 hours for spin up and 2 hours for registration with 30 s intervals. The trace gas emission rate is set as 1kg s$^{-1}$. The simulated concentration is scalable with source emission rates, as simulated by passive trace gas dispersion.

The simulated 3D volume concentration snapshots are then integrated by each column weighted by column averaging kernel (Bovensmann et al., 2010; Jongaramrungruang et al., 2019). The column averaging kernel is a vector representing the vertical sensitivity distribution of the instrument and retrieval algorithm, and it is here considered to be vertically uninformed. The resulting 2D column mass field is then subjected to additive Gaussian noise, considering instrument and retrieval uncertainty.





The noise is given as a percentage of methane's mean dry column concentration, which is considered 1.8 ppm (i.e., $\approx 10.3$ g m$^{-2}$ at 1 atm dry air).

### 2.2.2  Synthetic observations

To evaluate the possible impact of plume overlapping on quantification and the performance of the proposed separation method, we performed observation system simulation experiments (OSSEs) with simulated mass columns by WRF-LES. The OSSEs
are widely applied to evaluate the spaceborne GHG source detection and quantification abilities by simulating observed spectral radiations or retrieved concentrations (Bovensmann et al., 2010; Kuhlmann et al., 2019; Varon et al., 2018). OSSEs with realistic LES simulations, accounting for actual surface topology and meteorological conditions (Stoll et al., 2020), are preferable for specific source targets, however, the computational cost can be expensive, considering massive point source targets with highly heterogeneous spatial and emission conditions, e.g., targets in Duren et al. (2019). One feasible approach is to sum
multiple simulated column mass images after rotations, shiftings and concentration scalings while assuming the turbulence variations between multiple mass images are minor and negligible for quantification algorithms. This approach allows for simulating sources with arbitrary emission rates, spatial and meteorological conditions, allowing much lower computation cost and thus reducing linear time complexity ($O(N)$) to nearly constant ($O(1)$) for simulating $N$ sources when $N$ is large enough.

To synthesize an observation with multiple sources, we establish an easting/northing Cartesian system where the x-axis
points east and the y-axis points north. The field of view (FoV) is a square with sides at the length of 6 km and parallel to the axes, centred at $(0,0)$ of the Cartesian system. Then, $N$ of 2D column mass snapshots are randomly selected with the given wind speed and mixing depth. These snapshots are then scaled according to the emission rate of each source. Then, to rotate and shift the snapshots for an observation, we traverse all the pixels in the observation and accumulate their mapping pixels in each snapshot. For a given pixel of the observation at $(x, y)$, the mapping pixel indexing in the $n$th snapshot is given by

$$
\begin{bmatrix} i' \\ j' \end{bmatrix} = \begin{bmatrix} \cos\theta & \sin\theta \\ -\sin\theta & \cos\theta \end{bmatrix} \begin{bmatrix} x - x_n \\ y - y_n \end{bmatrix} \cdot \begin{bmatrix} \frac{1}{\Delta x} \\ \frac{1}{\Delta y} \end{bmatrix} + \begin{bmatrix} I_{\text{source}} \\ J_{\text{source}} \end{bmatrix}, \tag{12}
$$

where $\theta$ represents the wind angle to a-axis; $(x_n, y_n)$ represents the location of source $n$; $\Delta x$ and $\Delta y$ represent the horizontal resolutions in WRF-LES; $(I_{\text{source}}, J_{\text{source}})$ represent the location of the source pixel in WRF-LES. The indices $(i, j)$ are then rounded, thus completing the nearest neighbour interpolation. Here, considering the small size of the plume and the domain, we adopt a unified wind velocity across the domain.

### 2.2.3  Observation scenarios

We test our separation method under three different scenarios, namely Exp1-3, each consisting of trials with various experiment settings. Typical cases of each scenario are shown in 2. Exp1, the single source scenario, comprises a full factorial experiment of environmental factors, source factors and observation factors to evaluate the performance of quantifying methods on the single source. Exp2, the dual source scenario, comprises a full factorial experiment with several overlapping related factors



to evaluate the performance of separation and quantification methods. Exp3, the random source scenario, comprises a Monte
       Carlo test to further evaluate the separation and quantification methods.

       In Exp1, a single source is placed in the centre of the simulation domain and a full factorial experiment is conducted to
       test the performance of the quantifying method under all combinations of levels of multiple factors. These factors include
       environmental factors (mixing depth at 500, 800, and 1100 m; wind speed at 1, 3, 5, 7, 9 m s$^{-1}$; wind direction at 0, 45°),
source factor (emission rates ranging from 100 to 2000 kg h$^{-1}$), and observation factors (ground pixel size ranging from 25 to
       200 m; retrieval uncertainty at 1, 3, and 5%). The wind direction is defined in the Cartesian coordinate system, and the retrieval
       uncertainty is considered a 0-biased additive noise with standard deviation as a percentage of methane's mean dry column
       mass. Each combination is repeated for 10 times. The quantification performance of Gaussian plume fitting, unseparated IME
       (UNSEP) and separated IME (SEP) methods are evaluated. The results are elaborated in Section 3.1.

In Exp2, a secondary source is introduced as an interference source to produce overlapping, as to evaluate the impact on
       quantification and separation. Exp2 is also a full factorial experiment, where we fix the mixing depth at 800 m, the emission
       rates at 200 kg h$^{-1}$, the ground pixel size at 25 m, and the retrieval uncertainty at 1%. The rest factors include wind speed at 1, 3,
       5, 7, 9 m s$^{-1}$; wind direction ranging from -90 to 90°; distance between ranging from 100 to 900 m; and the emission rates ratio
       between the secondary and the original source ($Q_2/Q_1$) ranging from 0 to 5. Each combination of trials is repeated 10 times.
The quantification performance of single source Gaussian plume fitting, multi-source Gaussian plume fitting, unseparated IME
       and separated IME methods are evaluated. The results are elaborated in Section 3.2.

       In Exp3, a Monte Carlo experiment is conducted to further assess the performance of the unseparated IME and separated
       IME methods. For each trial, one source is randomly sampled from the AVRIS-NG observed methane source list (Duren et al.,
       2019) and its geolocation and emission rate is thus specified. Likewise, additional neighbouring sources within the 6 km ×
6 km domain are then included in the simulation. Sources with emission rates lower than 25 kg h$^{-1}$ (accounting for about
       5% of the summation) are excluded from the list as they are considered too small to be accurately measured by spaceborne
       measurements. The persistence of sources is assumed to be 1, as to demonstrate an aggressive estimate on plume overlapping.
       The wind velocity is then obtained from the fifth generation of atmospheric reanalysis of the European Centre for Medium-
       Range Weather Forecasts (ECMWF-ERA5; Hersbach et al., 2020). The sampling time range for loading wind velocity covers
the entire local noon in 2022. The wind velocity is considered uniform across the domain and is interpolated to the observation
       center using 5-point inverse distance weighting such as Xu et al. (2022). To ensure that all generated source inside the domain is
       quantifiable, each side of the simulation frame is extended outward 2 km for a 10 km × 10 km domain. This random experiment
       is repeated 2000 times. The quantification performance of the unseparated IME and separated IME methods is then evaluated
       and elaborated in Section 3.4.

In this study, the integrated mass enhancement (IME) method is adopted as a representative of the single source quantification
       methods. The emission rates estimated by IME method is given by Varon et al. (2018)

$$Q = \frac{U_{\text{eff}} \cdot \text{IME}}{L} = \frac{U_{\text{eff}} \cdot \sum_{(x,y) \in \mathbf{I}} C(x,y) A(x,y)}{\sqrt{\sum_{(x,y) \in \mathbf{I}} A(x,y)}}, \tag{13}$$





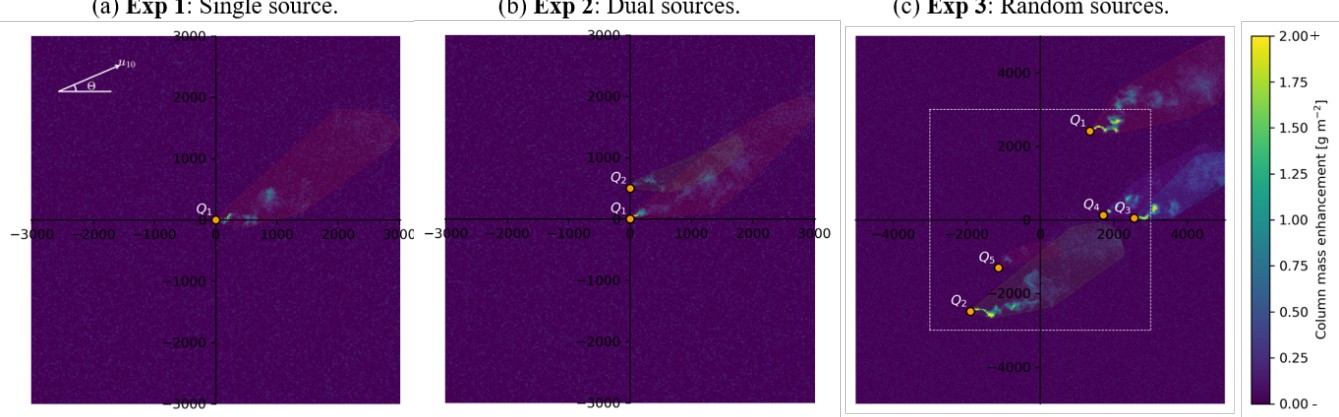

**Figure 2.** Typical cases for each experiment scenario of synthesized observation. The plots represent the synthesized methane column observation for evaluation. Each semi-transparent polygon patch covers a plume, where the concentration is larger than the uncertainty. The dashed box in (c) encloses the area where random sources are generated.

where $\mathbf{I}$ represents the set of pixels identified as plume, $C(x,y)$ represents the mass enhancement of pixel $(x,y)$, $A(x,y)$ represents the area of pixel $(x,y)$. The effective wind speed is a logarithmic functions with linear variations to 10-m wind speed $U_{10}$, where the parameters are fitted using the WRF-LES simulations (Varon et al., 2018). Our fit result is $U_{\text{eff}} = 0.55 \log U_{10} + 0.62$.

To enable the automatic application of the IME method to multi-source observations, firstly, the plume detection algorithm by Varon et al. (2018) is utilized to detect plume pixels. Then, pixel connectivity analysis, a morphological image processing technique, is applied to attribute the nearest connected structure to each source. The pixels inside these connected structures are then applied to the IME quantification.

## 2.3 EMIT observation

We also test our separation method on methane plumes retrieved by the Earth Surface Mineral Dust Source Investigation (EMIT) instrument installed on the International Space Station (ISS). The EMIT is a hyperspectral instrument capable of imaging spectroscopy in the visible to short wavelength infrared, with a nadir ground sampling distance of 30-80 m.(Green et al., 2020). The methane column enhancement data, labeled(EMITL2BCH4ENHv001) is expressed in units of parts per million meters (ppm m) and is retrieved using an adaptive matched filter technique. (Green et al., 2023b). Green et al. (2023a) also provides the corresponding identified plume complexes (EMITL2BCH4PLM v001), where the plumes sometimes overlap and thus form these clusters.

For the quantification, we first convert the concentration map from ppm m to kg m$^{-2}$ (Sánchez-García et al., 2022). Then, we applied the separation method described in Section 2.1 to extract plumes from each source. The extracted plumes are then quantified by the IME method. The wind velocity for separation and quantification is interpolated from ERA5 as described





in Section 2.2.3. The source locations are identified through visual inspection, and cross-verified with local ground facilities using high-resolution satellite map from Google Earth. Monte Carlo propagation is introduced to evaluate the uncertainty of the quantification (Sánchez-García et al., 2022). The input uncertainties include observation uncertainty from the corresponding

EMITL2BCH4PLM data, and wind speed uncertainty estimated as the standard deviation of the nearest 5 points from ERA5. The accuracy of IME method may also be subjected to multiple factors, including retrieval uncertainty, background, as well as the the pixel detection method (Jongaramrungruang et al., 2019; Duren et al., 2019). So the systematic errors of IME method is considered beyond the scope of this work.

### 2.4 Evaluation indicators

#### 2.4.1 Overlapping indicator

To assess the degree of plume overlapping, a mass overlapping index is proposed, defined as the ratio of the integration of mass contributions from the primary source and other sources to the mass of the primary source plume. The mass overlapping index for source $i$ of $N$ sources is given by

$$\mathrm{OI}_{\mathrm{mass}_i} = \frac{\sum_{(x,y)\in I}[(\sum_{n=1}^{N} C_n(x,y) - C_i(x,y)) \cdot A(x,y)]}{\sum_{(x,y)\in I} C_i(x,y)A(x,y)} \tag{14}$$

where $I$ denotes the plume pixel of source $i$. Higher $\mathrm{OI}_{\mathrm{mass}_i}$ denotes that the plume of source $i$ is subject to more severe interference.

#### 2.4.2 Emission rates estimation indicators

The quantification of methane source is considered as solving a regression problem. So the $R^2$, coefficient of determination, is introduced to indicate the accuracy of overall regression results. Furthermore, as $R^2$ has a relatively poor ability to explain

regression results with small true values, absolute percentage error (APE) is introduced to indicate the regression error of a single sample, and mean absolute percentage error (MAPE) is introduced to indicate the overall regression error. The definitions of $R^2$, APE and MAPE are given by

$$R^2 = 1 - \frac{\sum_{n=1}^{N}(\hat{Q}_n - Q_n)^2}{\sum_{n=1}^{N}(\bar{Q} - Q_n)^2}, \tag{15}$$

$$\mathrm{APE}_n = \left| \frac{\hat{Q}_n - Q_n}{Q_n} \right|, \tag{16}$$

$$\mathrm{MAPE} = \frac{1}{N} \sum_{n=1}^{N} \left| \frac{\hat{Q}_n - Q_n}{Q_n} \right|, \tag{17}$$

respectively, where $Q_n$ and $\hat{Q}_n$ represent the true emission rate and predicted emission rate, respectively, of source $n$; $\bar{Q}$ is the average of true emission rates; $N$ represents the number of sources in the experiments.



## 3 Results

### 3.1 Quantification results on single source

In Exp1, we evaluate the baseline performance of various quantification methods, including the emission rates derived directly from the Gaussian plume fitting, unseparated and direct IME quantification (denoted as UNSEP), and quantification after applying the separation method for single source for the extraction (denoted as SEP), using a full factorial experiment.

The overall quantification errors (MAPE) for the three quantification methods are 0.89, 0.30, and 0.40, respectively. The distributions of quantification errors, in terms of absolute percentage error (APE), with respect to different simulation factors in Exp1, are shown in Fig. 3.

As shown in Fig. 3(a), the APE of all three methods exhit nearly linear incresing trends with respect to pixel size, with Pearson's correlation coefficients ($R$) of 0.24, 0.18 and 0.21, and all p-values are less than 0.01. Similar trends are also shown in Fig. 3(b), where the APE of three methods increase slightly with respect to uncertainty ($R = 0.16, 0.13, 0.14$; $p < 0.01$). As shown in Fig. 3(c) and Fig. 3(d), the variance of APE with respect to mixed depth and wind direction is minor. As shown in Fig. 3(c) and Fig. 3(d), the variances of APEs with respect to mixed depth and wind direction are minor. As shown in 3(e), the APE of Gaussian plume and UNSEP increase with respect to the wind speed. However, the APE of the SEP reaches the maximum at the wind speed of 3 m s$^{-1}$. With increasing wind speed, SEP exhibits lower quantification error than UNSEP in 9 m/s. As shown in Fig. 3(f), the quantification error of all three method decrease with the emission rates, and shows a sub-linear trend.

### 3.2 Quantification results on dual sources

In Exp2, we introduce an interference source as to create overlapping for the full factorial experiment. After introducing an interference source, the MAPE of single source Gaussian plume fitting increases from 0.45 to 1.23, while the increases of multi-source Gaussian plume model are negligible, which remains 0.45. Similarly, the UNSEP increases from 0.15 to 0.83, while the SEP only increases from 0.30 to 0.38.

As demonstrated in Fig. 4, the SEP shows the best quantification performance in most cases, followed by multi-source Gaussian plume fitting, UNSEP and single-source Gaussian plume fitting. With decreasing wind speed, the errors of quantification results by multi-source Gaussian plume fitting become comparable to that of SEP. When the wind speed is 1 m s$^{-1}$, the quantification results of multi-source Gaussian plume fitting demonstrate slight superiority over SEP. With increasing distance and interference strength, where plumes are less likely to overlap, the performance of UNSEP begins to demonstrate superiority. When the distance increases to 900 m and the interference decreases to 0.5, the SEP shows the best quantification performance.

Additionally, the multiple source Gaussian plume and SEP exhibit better quantification performance on overlapping plumes as interference strength $Q_2/Q_1$ intensifies. Besides, both multi-source Gaussian plume fitting and SEP show minor variations over factors including wind direction, wind speed, distance between two sources and interference strength $Q_2/Q_1$. In comparison, the quantification results of traditional single-source Gaussian plume fitting and UNSEP are more susceptible to the listed factors. Their performances deteriorate as wind direction aligns increasingly with the line connecting the two sources,





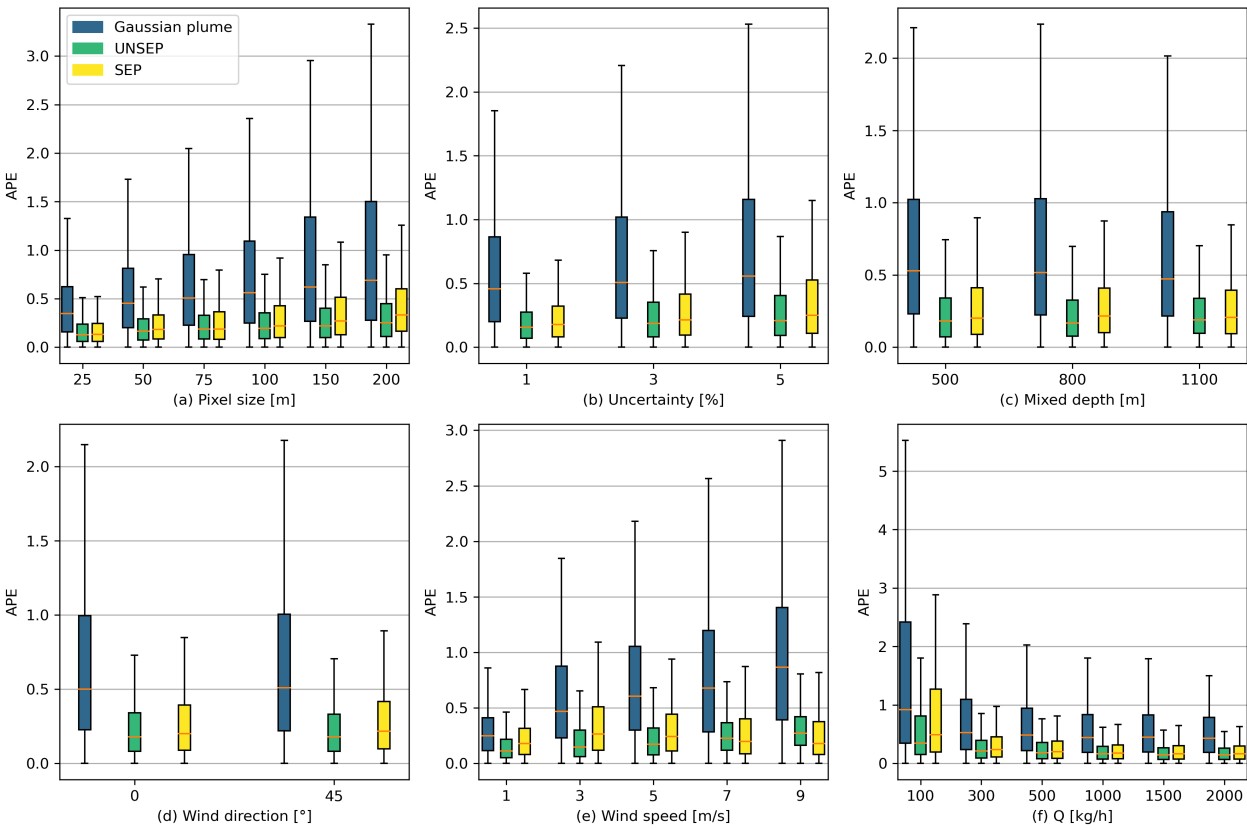

**Figure 3.** Distribution of quantification APE under various experimental parameters in EXP1. The orange dashes denote the medians of APE; the boxes denote the range between the lower and upper quartiles (Q1 and Q3); ⊥ and ⊤ extend from the box by 1.5 times the inter-quartile range (IQR). The quantification errors in APE of Gaussian plume fitting, unseparated IME (UNSEP) and separated IME (SEP) methods are represented with legend.

decreasing or increasing wind speed (for UNSEP and single source Gaussian plume fitting, respectively), closer distance and stronger interference source.

## 3.3 Comparison between RMS and RMS+LBPM as the minimization objective

To evaluate the improvement of the proposed multi-objective optimization (Eq.6, denoted as RMS+LBPM) compared to RMS-minimizing optimization (Eq.5) on quantifying relatively weaker source, experiments are carried out based on Exp2, only appending interference strength $Q_2/Q_1$ with $\{10, 15, 20\}$ where higher $Q_2/Q_1$ means stronger interference to primary source of $Q_1$.

The results are shown in Table 1, MAPEs of both methods increase with the interference strength $Q_2/Q_1$, and the MAPE of RMS+LBPM is lower than the MAPE of RMS for each interference strength. The APE of RMS+LBPM is significantly lower





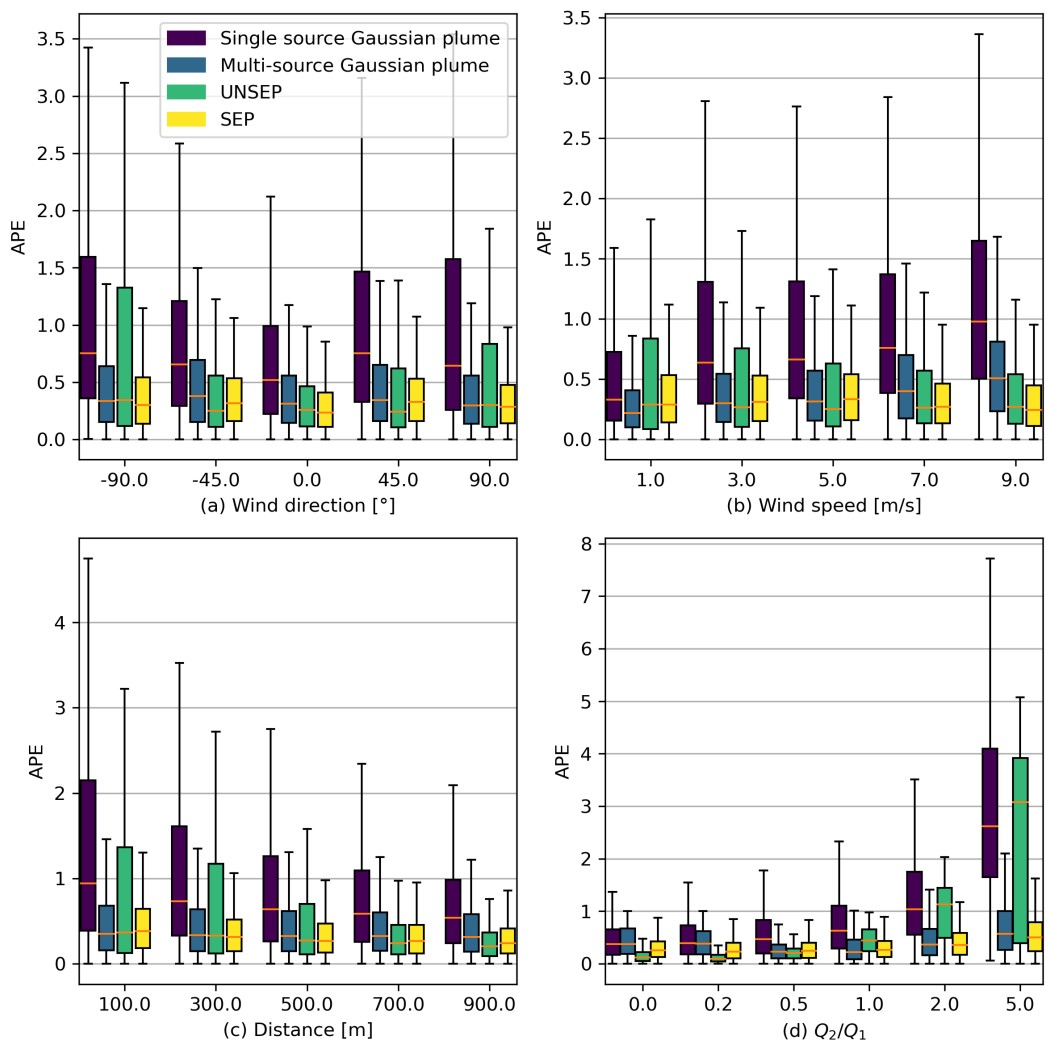

**Figure 4.** Distribution of quantification APE with respect to various experimental factors in EXP2.

($p < 0.1$ for t-test) when $Q_2/Q_1 \in \{1/5, 5/1, 10/1, 15/1\}$, where the discrepancy in emission rates is relatively large. This indicates that RMS+LBPM can serve as a slightly better minimization objective compared to RMS for separating overlapping plumes of sources with uneven emission rates.

## 3.4 Quantification results on random sources

In Exp3, we focus on comparing the UNSEP and SEP in a more realistic Monte Carlo simulation. Factors including source locations, emission rates and wind velocities are randomly selected from real distributions. The sampled factors demonstrate good agreement in terms of source emission rates (Fig. 5(a)), and wind speed (Fig. 5(b)) with the real distributions of the entire





**Table 1.** Comparison between RMS and RMS+LBPM as the Minimization Objective.

| $Q_2/Q_1$ | MAPE of RMS* | MAPE of LBPM+RMS** | $p$ |
|---|---|---|---|
| 1/5 | 0.42 | **0.41** | $p < 0.1$ |
| 1/2 | 0.47 | **0.45** | - |
| 1/1 | 0.54 | **0.52** | - |
| 2/1 | 0.61 | **0.60** | - |
| 5/1 | 0.65 | **0.62** | $p < 0.05$ |
| 10/1 | 0.72 | **0.68** | $p < 0.1$ |
| 15/1 | 0.85 | **0.81** | $p < 0.1$ |
| 20/1 | 0.94 | **0.92** | - |

* RMS: Estimation with $\mathcal{L}_{\mathrm{RMS}}$ as the minimization objective (see Eq. 5).

** LBPM+RMS: Estimation with multi-objective optimization (see Eq.6).

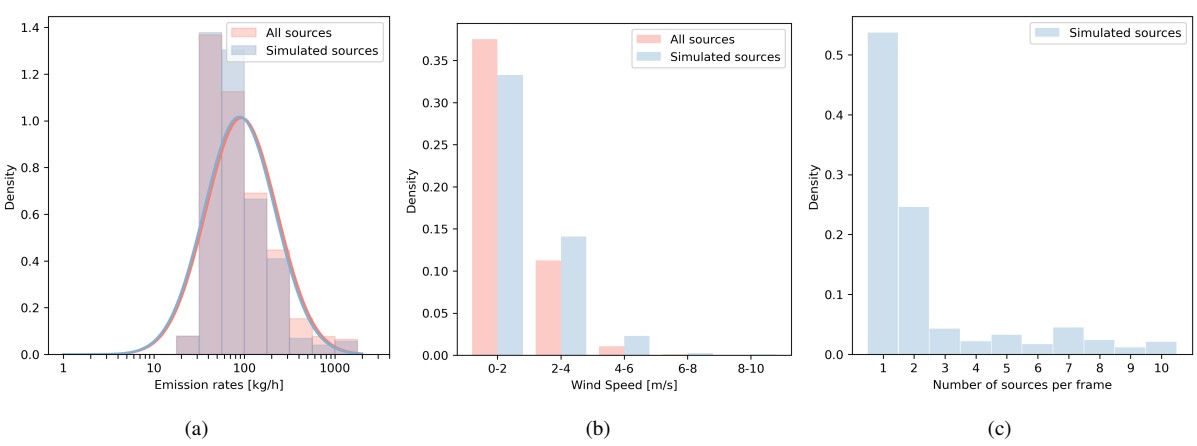

(a)          (b)          (c)

**Figure 5.** Statistical description of the factors in the Monte Carlo experiment (Exp3). ▉ denotes the distribution of all sources from AVIRIS-NG methane source inventory and the corresponding local noon wind speed distribution in the entire year of 2022. ▉ denotes the distribution of the selected sources in EXP3.

source list. The sampled emission rates follow a log-normal distribution, with mean of 172.2 kg s$^{-1}$ and standard deviation of
390   340.8 kg s$^{-1}$. As shown in Fig. 5(c), 53.7% of the frames cover one source, 24.6% of the frames cover two sources, and 21.7% of the frames cover more than two sources. On average, there are 2.33 sources per frame (6 km $\times$ 6 km).

Fig. 6 shows the quantification results of unseparated and separated IME quantifications. It is observed there is a linear underestimation for UNSEP (Pearson's $R = 0.93$, $p < 0.01$). The SEP estimations are then corrected with regression result





$y = 0.51x - 1.31$, where $x$ represents the corrected emission rates and $y$ represents the underestimated emission rates. After
correction, the SEP shows an improvement in $R^2$ from 0.71 to 0.84 compared to UNSEP, and a decrease in the quantification error (MAPE) from 1.46 to 0.45. We also find that SEP is notably more accurate in estimating of low-emission sources compared to UNSEP.

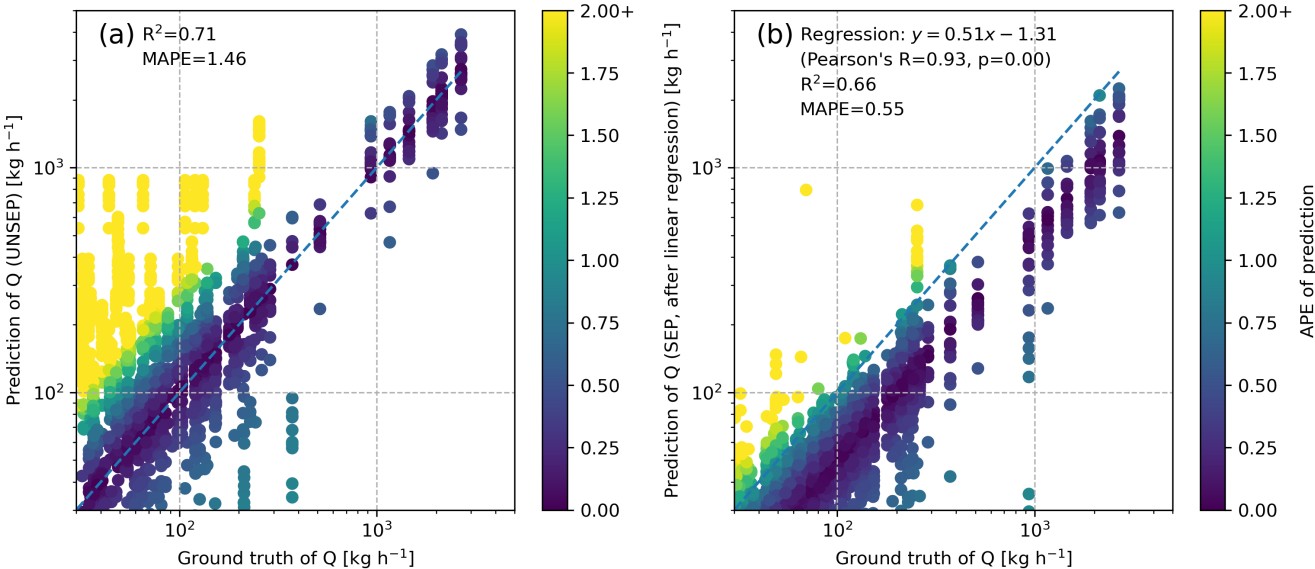

**Figure 6.** Comparison between quantification results of (a) unseparated quantification and (b) separated quantification in Exp3. The dash lines denote the 1:1 lines. Linear correction is applied to the SEP results in (b).

To further investigate the distribution of overlapping and the performance of UNSEP and SEP in handling overlapping, we demonstrate quantification error over the source overlapping index. The overlapping index $\mathrm{OI_{mass}}$ ranges from 0 to 6.09. Only 36.0% of sources are completely isolated from other sources and their overlapping index $\mathrm{OI_{mass}}$ is 0; half of the sources are with $\mathrm{OI_{mass}} > 0.02$; and 4.3% of sources are be subjected to overlapping with $\mathrm{OI_{mass}} \geq 1$. We observe a linear relation relationship between APE of UNSEP and $\mathrm{OI_{mass}}$ (Pearson's $R = 0.45$, $p < 0.01$), and the regression result can be expressed as $\mathrm{APE_{UNSEP}} = 2.76 \cdot \mathrm{OI_{mass}} + 1.34$. We define that when the APE exceeds 2 times as the intercept, the corresponding overlap situation is considered severe. Then, this results in a $\mathrm{OI_{mass}}$ threshold of 0.41, indicating that 18% of the sources are suffering from severe overlapping. In comparison, the corresponding $\mathrm{OI_{mass}}$ threshold of SEP is 16.58, indicating the effect of overlapping is largely depressed and thus results in robust quantification.

### 3.5 Quantification on EMIT observation

In this section, we validate our separation method using real satellite observation of EMIT. We focus on a specified cluster of plumes observed by EMIT on 15th August 2022 at 4:28 (UTC) in Turkmenistan, near the Goturdepe oil and gas production field. The sources in this location are spatially aggregated and create significant plume overlapping (see Fig. 8). Through manual





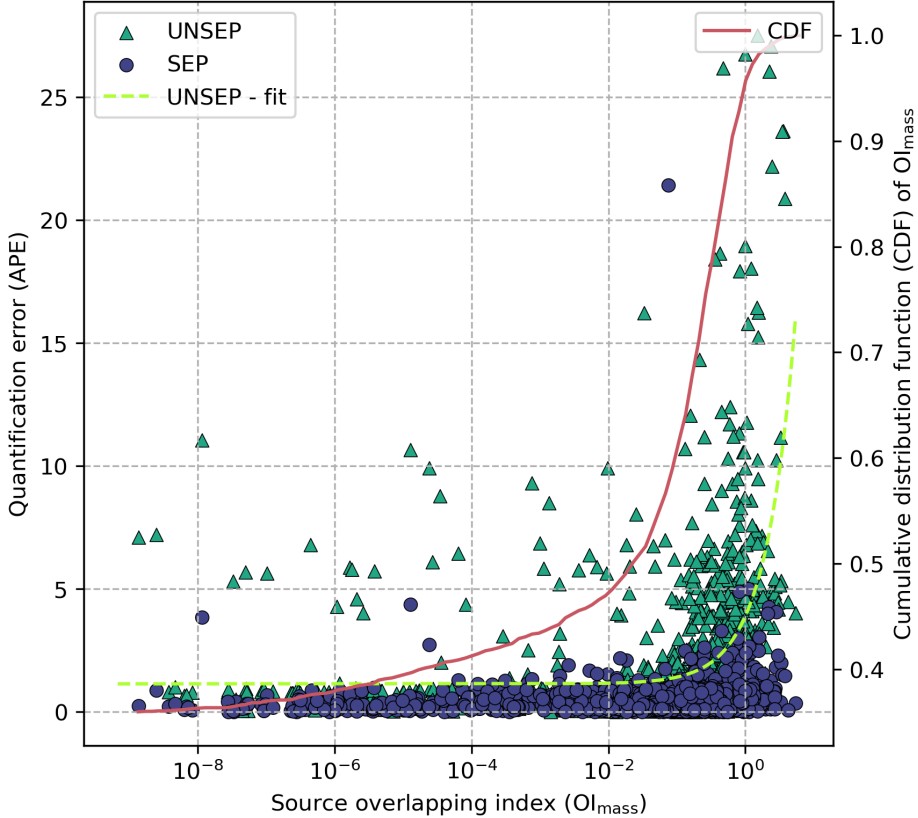

**Figure 7.** Comparison between quantification results and source overlapping index ($OI_{mass}$). The dash-line represents the linear fitted quantification error of UNSEP with respect to $OI_{mass}$. The red solid line represents the cumulative distribution function of $OI_{mass}$.

inspection and high-resolution satellite imagery verification, we identify 6 sources within the cluster. The emission rates of each source is quantified using our separated quantification method. Additionally, we quantify the entire cluster as a whole using the conventional unseparated IME method, which does not include separation, as well as the connectivity verification.

The quantification results are shown in Table 2. The estimated emission rates for each source range from 1.64 to 5.20 t

h$^{-1}$. We compare our estimated emission rates with previous research. We find that source $Q_3$ has also been quantified by Irakulis-Loitxate et al. (2022) and Sánchez-García et al. (2022) and their estimations for $Q_3$ are $1.4 \pm 0.4$ t h$^{-1}$ and $5.0 \pm 2.2$ t h$^{-1}$, respectively.There is a gap of more than two years between these two estimations, and their estimations demonstrate significant difference. Our estimation for $Q_3$ on 15th August 2022 is $3.34 \pm 0.90$ t h$^{-1}$, which is comparable to the previous estimations. The summation of separated quantification results on the 6 sources is $16.77 \pm 4.65$ t h$^{-1}$. In comparison, the

quantification result of the whole cluster is $21.06 \pm 5.51$ t h$^{-1}$, which is higher than the summation but the their difference are consistent within margins of error. It's reasonable as pixels in separated quantification may not be attributed to any source and thus excluded in the final quantification, leading to underestimations.





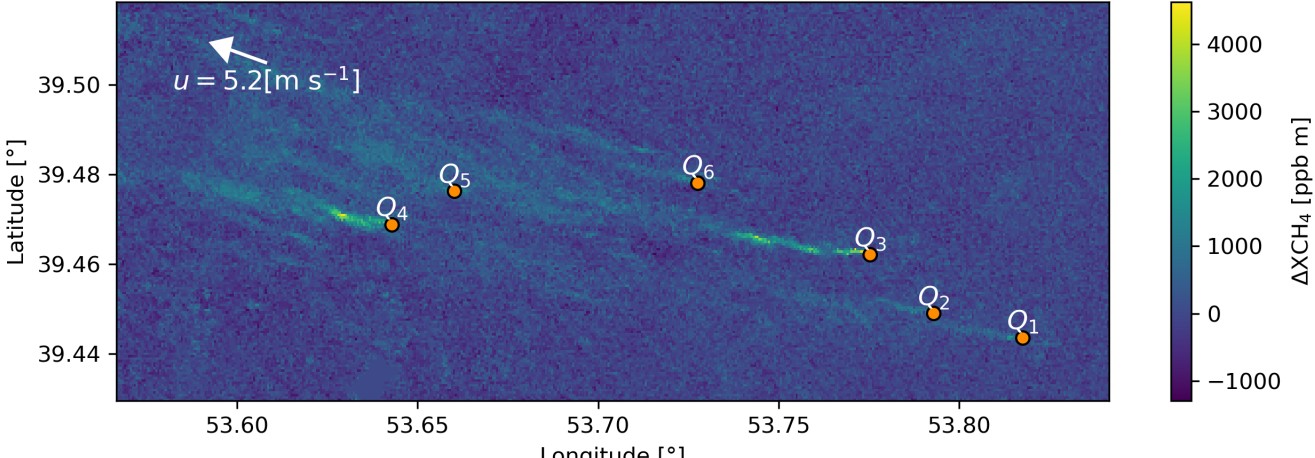

**Figure 8.** Real satellite observed methane column enhancement by EMIT on 15 August 2022 at 4:28 (UTC). This image is from the dataset EMITL2BCH4ENH v001, which is publicly available at https://lpdaac.usgs.gov/products/emitl2bch4enhv001/.

**Table 2.** Quantification results of EMIT observation. 6 sources are manually identified and quantified. The quantification results summation of separated IME method over each source separately (Summation) is compared to the results of unseparated IME method over the whole methane plume cluster (Whole).

| Source ID | Source location | Estimated emission rates [t h$^{-1}$] | Reference emission rates [t h$^{-1}$] |
|:---:|:---:|:---:|:---:|
| $Q_1$ | (39.4436°N, 53.8176°E) | $1.64 \pm 0.49$ | |
| $Q_2$ | (39.4490°N, 53.7929°E) | $1.35 \pm 0.42$ | |
| $Q_3$ | (39.4620°N, 53.7753°E) | $3.34 \pm 0.90$ | $1.4 \pm 0.4$ [*] |
| | | | $5.0 \pm 2.2$ [**] |
| $Q_4$ | (39.4687°N, 53.6428°E) | $5.20 \pm 1.38$ | |
| $Q_5$ | (39.4762°N, 53.6602°E) | $1.69 \pm 0.50$ | |
| $Q_6$ | (39.4781°N, 53.7276°E) | $3.55 \pm 0.95$ | |
| Summation | | $16.77 \pm 4.65$ | |
| Whole | | $21.06 \pm 5.51$ | |

[*] Observed by PRISMA on 27 March 2020 (Irakulis-Loitxate et al., 2022).

[**] Observed by WorldView-3 on 10 April 2022 (Sánchez-García et al., 2022).

## 4 Discussion and conclusions

In this study, we investigated the impact of plume overlapping on spaceborne methane point source monitoring. We found that
plume overlapping increases quantification errors in Exp2 for IME quantification, where MAPE increases from 0.15 to 0.83





compared to compared to no interference cases. Factors such as closer source intervals and disproportion emission rates will enlarge these defects. Overlapping plumes can produce connective pixels that cover multiple sources, which are ambiguous to be attributed. Simply eliminating these pixels will result in increasing missed detections and quantification errors. In addition, the relatively sparse spatial resolution of spaceborne methane monitoring techniques compared to airborne techniques can

increase the proportion of these ambiguous pixels. As a result, it's essential to find and attribute pixels in overlapping plumes correctly for spaceborne quantification .

To tackle this issue, we introduced a multi-objective heuristic optimization algorithm to perform parameter estimations for the 2D multi-source Gaussian plume model. Based on the outputs of this model with the estimated parameters, we assigned the mass to sources according to the modelled concentrations by each pixel. In this way, we separated an overlapping-plume image

into several single-plume images. This "soft segmentation" shows better performance in plume pixel detection on overlapping plumes than "hard segmentation" methods, e.g., the plume detection method by Varon et al. (2018), which assigns all the mass in a pixel to a single source while the mass may originate from multiple sources. Results in the Monte Carlo experiment show that the application of separation is effective in quantification, where MAPE decreases from 1.46 to 0.45, and $R^2$ increases from 0.71 to 0.84, compared to quantification without separation.

Additionally, our separation model can perform independent estimation for attributes such as wind speed and direction, and source locations, which make the separation robust to the prior uncertainty of these factors. Although the emission rates as parameters of the 2D multi-source Gaussian plume model are estimated, they are only used for separation instead of quantification. As our experiment results in Exp1 and Exp2 shows, Gaussian plume fitting exhibit higher systematical uncertainty than the IME method on quantifying fine-scale methane plumes. It need to be noticed that, in our experiment, the quantification

error of Gaussian plume increase with pixel size and is constantly higher than the IME methods as shown in Exp1. It is slightly different from Varon et al. (2018); Jongaramrungruang et al. (2019), which shows that plume shows better approximation of Gaussian form with increasing pixel size above 300 meters. One explaination is that our WRF-LES simulation domains are samll, and such scale may not help averaging the eddies. A comprehensive exploration of the trade-offs between Gaussian plume and IME methods may require large-scale, high-resolution LES simulations, and it is beyond the scope of this paper.

However, the Gaussian plume model is still statistically correct, which means it still can be used to perform rough estimations (e.g., Jacob et al., 2022). Acknowledging this fact, we also apply the Gaussian convolution kernel to blur the modelled plumes to increase the robustness in capturing the plume structure.

In the experiment on real satellite observations, firstly, we notice that identifying source locations correctly is crucial for separation and quantification. Although we verified the source with satellite imagery, it still appears less precise. Utilizing detailed

facility-level inventories, such as VISTA-CA, could greatly help source detection, separation and quantification. Additionally, for previously unknown emission sources, integrating multimodal information, including pipeline maps and simultaneous facility flare observations can also be introduced for accurate identifying (Irakulis-Loitxate et al., 2022). Secondly, we also notice that there are unignorable differences in source quantification results across research, as shown in Section 3.5. This suggests further validation to study the systematic errors of the IME quantification method. Thirdly, we find that in some cases the

plumes exhibit a large deviation from the WRF-LES simulation, especially in complex terrains, such as valleys. In this case,

using uniform wind assumptions may also lead to the overestimation of the performance of the IME quantification method as well as the separation method.

In this study, we proposed a method for separating the overlapping plumes from multiple facility-level point sources in spaceborne methane observations, thus extending the conventional single-source quantification methods to be applicable under plume-overlapping observations. As implied by the VISTA-CA inventory and AVIRIS-NG observed methane source list, the methane point sources can be spatially aggregated in some places, meaning that the plume overlapping may be non-negligible. This demerit will constrain the quantification scope of spaceborne GHG monitoring techniques. As a result, our separation method can be important to spaceborne methane monitoring for constructing or verifying facility-level emission inventories (e.g., Duren et al., 2019), as well as environmental administration departments. For future research, a dispersion model, which is more representative of real transient plumes, can be introduced to improve the separation performance. As our separation method is easily cascadable, new plume pixel detection methods can be introduced for better separation. A series of tests on more realistic simulations, as well as real observations, should also be performed for further validation.

*Code availability.* The original version of the WRF source code is publicly available at https://www.mmm.ucar.edu/models/wrf; the source code of the proposed separation method in Section 2.1 is available upon request.

*Data availability.* The VISTA-CA inventory is publicly available at https://doi.org/10.3334/ORNLDAAC/1726; the AVIRIS-NG observed methane source list is publicly available as the supplementary information at https://doi.org/10.1038/s41586-019-1720-3; the ECMWF-ERA5 reanalysis meteorological data are publicly available at https://www.ecmwf.int/en/forecasts/dataset/ecmwf-reanalysis-v5; the EMIT methane enhancement data are publicly available at https://lpdaac.usgs.gov/products/emitl2bch4enhv001/; the EMIT estimated methane plume complexes are publicly available at https://lpdaac.usgs.gov/products/emitl2bch4plmv001/.

*Author contributions.* YP designed and implemented the study, as well as composed the draft. GL, LT and SG conceptualized the objective of this study. DH and GL reviewed and edited the manuscript. All authors reviewed the manuscript.

*Competing interests.* The author has declared that there are no competing interests.

*Financial support.* This research has been supported by the National Key Research and Development Program of China (grant no.2022YFB3904800).



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
