# Peer review of "Separating and Quantifying Facility-Level Methane Emissions with Overlapping Plumes for Spaceborne Methane Monitoring"

_EGUsphere, 2023_

## Author Comment (AC1)

**Author's response to RC1**

Dear Referee,

Thank you for your insightful comment. I will present our point-by-point feedback as follows.

**Point-by-Point Feedback**

- *"The overlapping of plumes is a real scenario that can occur worldwide. However, it is not understood the impact of these cases at a global level. Thus, it could be a very specific topic that is only applicable in some examples."*

**Response:** Thanks for your advice. Our motivation for separating methane plumes for quantification has arisen from real-world cases, e.g., the recently published research by EMIT (Thorpe et al., 2023). They manually inspected the L2 data and found plume overlapping can be common. The plume overlapping is especially common in O&G concentrated areas, which generally agrees with our analysis on VISTA-CA source inventory, as well as our Monte Carlo simulation for AVIRIS-NG observed sources.

[Figure]

Figure RC1.1 Overlapping plumes observed by (a) Worldview-3 (Sánchez-García et al., 2022); (b) AVIRIS-NG (Duren et al., 2019); (c) EMIT (Green et al., 2023).

   While a comprehensive global analysis of plume overlapping appears essential, it can be challenging due to its dependence on both instrumental and environmental factors, in addition to the source distribution. Thus, it seems beyond the scope of this paper. Future work could further investigate this phenomenon with real satellite observations for deeper understanding.

- *"Moreover, the proposed methodology is rather complex. Is it possible to just quantify the non-overlapped area of the plumes (of e.g. FIg 2 and 8) and apply (considering the caveats) the IME method over them?"*

**Response:** While quantifying the non-overlapped area of the plumes using direct morphological methods seems an interesting possibility, it presents challenges. This method requires predefined shape priors of each plume to eliminate the overlapping area and attribute plumes to sources, which can be challenging. It usually requires massive manual inspection and labeling work, and may also introduce extra systematic errors caused by the subjectivity of the operators.

Even if an algorithm were implemented to automatically eliminate the overlapping pixels and perform source attribution, obstacles may occur in specific cases. (1) When a source is located downwind of an interference source and its plume is fully covered by the interference plume, the method above is not applicable. (2) In cases where the isolated pixels are near the source (in most cases), the quantification can be unstable. As shown in Figure RC1.2, the downwind IME near the source exhibits instability, and this uncertainty will propagate proportionally into the quantification results.

[Figure]

Figure RC1.2 The integrated mass enhancement (IME) distribution downwind in our simulation.

The IME is calculated by integration across wind direction (i.e., over the y-axis).

- *"The results indicate a strong underestimation in the quantified flux rate when applying the separation methodology. The authors argue in L421 that this could be the result of some pixels not being attributed to any source. Thus, the results indicate that the methodology needs to be reviewed."*

**Response:** We found an underestimation trend (see Figure 6), which exhibits a linear relationship with a correlation coefficient of 0.93 (R=0.93). Consequently, this underestimation is corrected using linear regression analysis.

We derived the correction coefficient from the simulation data and applied it to correct the quantification result of the EMIT observation. However, despite this correction, the separated quantification result on EMIT is still lower than the result of directly applying IME quantification, as shown in Table 2.

By superimposing plume pixels detected by the reference method (Varon et al., 2018; UNSEP) with those identified using our method (combination of separation and the reference method; SEP), we can see some plume pixels are excluded by our method (see Figure RC1.3). These excluded pixels can't be attributed to any source. This issue can be raised by irregular plume shapes, mislabeling of source(s), and interference from background noise. These challenges are common in this field. A detailed comprehensive discussion of these issues seems beyond the scope of this article.

[Figure]

Figure RC1.3 Whole plume pixels (yellow) and separated & attributed plumes (Viridis coloring).

**Reference**

- Duren, R. M., Thorpe, A. K., Foster, K. T., Rafiq, T., Hopkins, F. M., Yadav, V., Bue, B. D., Thompson, D. R., Conley, S., Colombi, N. K., Frankenberg, C., McCubbin, I. B., Eastwood, M. L., Falk, M., Herner, J. D., Croes, B. E., Green, R. O., & Miller, C. E. (2019). California's methane super-emitters. Nature, 575(7781), Article 7781. https://doi.org/10.1038/s41586-019-1720-3

- Green, R., Thorpe, A., Brodrick, P., Chadwick, D., Elder, C., Villanueva-Weeks, C., Fahlen, J., Coleman, R. W., Jensen, D., Olsen-Duvall, W., Lundeen, S., Lopez, A., & Thompson, D. (2023). EMIT L2B Methane Enhancement Data 60 m V001 [dataset]. NASA EOSDIS Land Processes Distributed Active Archive Center. https://doi.org/10.5067/EMIT/EMITL2BCH4ENH.001
- Sánchez-García, E., Gorroño, J., Irakulis-Loitxate, I., Varon, D. J., & Guanter, L. (2022). Mapping methane plumes at very high spatial resolution with the WorldView-3 satellite. Atmospheric Measurement Techniques, 15(6), Article 6. https://doi.org/10.5194/amt-15-1657-2022
- Thorpe, A. K., Green, R. O., Thompson, D. R., Brodrick, P. G., Chapman, J. W., Elder, C. D., Irakulis-Loitxate, I., Cusworth, D. H., Ayasse, A. K., Duren, R. M., Frankenberg, C., Guanter, L., Worden, J. R., Dennison, P. E., Roberts, D. A., Chadwick, K. D., Eastwood, M. L., Fahlen, J. E., & Miller, C. E. (2023). Attribution of individual methane and carbon dioxide emission sources using EMIT observations from space. Science Advances, 9(46), eadh2391. https://doi.org/10.1126/sciadv.adh2391
- Varon, D. J., Jacob, D. J., McKeever, J., Jervis, D., Durak, B. O. A., Xia, Y., & Huang, Y. (2018). Quantifying methane point sources from fine-scale satellite observations of atmospheric methane plumes. Atmospheric Measurement Techniques, 11(10), 5673–5686. https://doi.org/10.5194/amt-11-5673-2018

---

## Author Comment (AC2)

**Author's response to RC2**

Dear Referee,

Thank you for your insightful comment. We sincerely thank you for once again reviewing our manuscript carefully! I will present our point-by-point feedback as follows.

**Point-by-Point Feedback**

- *"The manuscript "Separating and Quantifying Facility-Level Methane Emissions with Overlapping Plumes for Spaceborne Methane Monitoring" by Y. Pang et al. documents an interesting topic and analysis regarding the quantification of pollutant or greenhouse gas emissions from industrial point sources based on satellite images. Such a study deserves a publication in AMT."*

**Response:** Thank you sincerely for recognizing our efforts. Your insightful and practical advice has helped greatly in the improvement of our manuscript.

- *"1) However, the text of the manuscript needs a major improvement and clarifications. a) There are signs of a lack of proofreading or of rigor in the writing, such as typos (e.g. at lines 39, 257, 273, 317, 351-352, 420, 426…), clumsy formulas (l.33; "given the same emission rate" at l35; "generally" at l.38, "the high nonlinearity of the Gaussian plume model" at line 78, "accurate priors" at l211, l214-215, "the increases of multi-source Gaussian plume model" at l359-360… ), misleading shortcuts (e.g. l41-42, l317-318…)."*

**Response:** We sincerely appreciate your pointing out our text shortcomings. These errors might have escaped our spelling and grammar checks, as well as our checks as authors for whom English is a second language. We will give it a multi-eyed and thorough check in the next revision, and try our best to avoid these grammatical errors.

- *"Some of the potential mistakes can be embarrassing: e.g. between mid-line 217 and line 218, I assume that the authors talk about chemistry rather than about*

*diffusion, and I do not really understand what the link can be between "the low concentration of methane in the atmosphere" and the diffusion or the chemistry. The word at line 237 is "topography" rather than "topology" ? etc."*

**Response:** Thank you for pointing out these potential mistakes. I will respond point-by-point as follows.

For, L217-L218: The original intent of the manuscript was that (on the simulation scale) tracer gas concentrations are low (compared to atmospheric background concentrations). For tracer gas transport, convective processes dominate the gas transport and the molecular diffusion term can be neglected. In the next revision, we will cite the opinion of Nottrott et al. (2014) directly here without a rationale.

For, L237: Thank you, it's a linguistic mistake. We will correct it in the next revision.

- *"b) In a general way, the concepts, plans and methods need to be better introduced and discussed."*

**Response:** Thank you for the suggestion. The point-by-point responses are as follows.

- *"From line 68, the introduction loses the clear distinction between the plume detection and the subsequent emission quantification when processing the CH4 plume images in two steps, or the distinction between such a two-step approach, and 1-step approaches such as Gaussian plume fitting (when using the emission rate from this fitting), while such distinctions are critical to follow the manuscript correctly."*

**Response:** Thank you for the suggestion. Although in some practices Gaussian plume fitting also requires manual specification of plume pixels and background pixels, explicit pixel detection algorithms are generally not emphasized in the related studies. If we perform fitting globally (e.g., the pixels in a surrounding rectangle), it can be considered as a "one-step" method. In comparison, other cases require a "two-step" detection-quantization approach.

However, we propose a different approach for the quantification of overlapping plumes. In this context, our proposed method can be considered as a "three-step" approach, i.e., separation-detection-quantification. We solve the overlapping plume

separation by fitting Gaussian plumes with a "one-step" method and use a specific "two-step" method to achieve more robust quantification compared to the "one-step" method.

We will make this part of the introduction more concise and clearer in the next revision.

- *"Lines 72-83 (which have been extracted from section 2) should be improved and better merged in the introduction."*

**Response:** Thank you for the suggestion. We will improve the text in the next revision.

- *"The abstract hardly manages to characterize the type of method that has been developed and tested in this manuscript: lines 3 to 7 seem to speak about a full quantification method and line 8 seems to introduce IME as a distinct benchmarking quantification method (and the rest of the abstract does not solve for this potential confusion)."*

**Response:** Thank you for the comments. As we mentioned above, we proposed a separation-detection-quantification schema. The detection-quantification method in the manuscript was basically modified from Varon's method (Varon et al., 2018), and is considered as representative detection-quantification method. However, theoretically, the detection-quantification method here can be replaced by other methods of the same purpose, e.g., a similar IME-based method (Duren et al., 2019), angular width method (Jongaramrungruang et al., 2019), even deep learning method (Jongaramrungruang et al., 2022).

For the coherence of the text, we will move this part to the discussion and make the text more clear in the abstract and introduction.

- *"I thank the authors for having added the results of the quantification of the emissions based on the Gaussian plume model fitting in the result section. However, now, this benchmarking needs to be announced in the introduction or early in section 2, this quantification method needs to be detailed in section 2, and the abstract should probably highlight the comparison between this quantification method and IME."*

**Response:** Thanks for the suggestion for comparing the results of our method and the direct Gaussian plume fitting. We will highlight it in the next revision.

- *"This is a significant result, which tend to confirm that the Gaussian plume model inversion does not behave as well (compared to IME) when tackling turbulent plumes at fine resolution than when tackling mesoscale plumes with > 1 km resolution images."*

**Response:** Our current findings show that the IME method outperforms Gaussian plume fitting in quantifying fine-scale plumes.

Limited by the computing constraints, we faced a resolution-scale trade-off in the WRF-LES simulation. Our preference for fine resolution limited the size of the simulation domain, preventing us from testing Gaussian plume fitting on larger scales. So, in Exp1, the number of plume pixels decreases with a sparser resolution, and the theoretical performance increase of the Gaussian plume with scale is not observed in the experiments.

- *"c) see the detailed comments below, which provide other illustrations of the general need to improve the text, its quality and its clarity, which applies to all sections."*

**Response:** Thanks for the detailed suggestions. We endeavor to improve the text in the next revision. Please see our point-by-point responses as follows.

- *"2) Section 4 makes an effort to synthesize the results from section 3, to provide some explanations for the behaviors of the methods as a function of the "experimental parameters". However, it could be extended and strengthened to better characterize the problematic and successful cases depending on the methods and to provide more interpretation."*

**Response:** Thanks for the suggestions and we will extend the discussion in the next revision. The following responses outline some areas for improvement.

- *"Furthermore, some items are probably missing such as a discussion on the impact of using the LBPM, and a discussion on the typical range of accuracy of the emissions estimates that can be expected as a function of the observation conditions."*

**Response:** Thanks for your suggestion, we will extend the discussion in the next discussion.

- *"Regarding the LBPM, could this approach be hampered by the retrieval noise, or by the regular transition to the RMS metric in the optimization iterative process ? See also some of the following comments that connect to potential gaps in this discussion section. Detailed comments:- is not the abstract misleading regarding the assessment of using the LBPM ? the results show that it does not impact much the results, while the abstract seems to say it is successful. Section 3.3 does not really highlight the fact that this impact is relatively small, and section 4 does not discuss it at all."*

**Response:** By definition (L196-197), the LBPM may be affected by strong noise ($\Delta C > 1\sigma$). The transition of LBPM to RMS primarily relies on empirical methods, as the stochastic nature of the differential evolution algorithm, makes it challenging to quantitatively analyze this iteration. In our preliminary trials, the total number of iterations generally ranges in the tens, with LBPM exponentially decaying to the same level as RMS within the first ~10 iterations.

The initial purpose of LBPM was to improve the sensitivity on shapes (gradient signs) for the fitting algorithm during the early iterations, thereby improving the fitting for weak point sources and avoiding local optima. LPBM performed exceptionally well in our preliminary tests on Gaussian plume synthetic observations. As illustrated in Figure RC2. 1, for fitting the synthetic observations (a), two estimates (b) and (c) were generated within an iteration. Despite a significant difference between its stronger source and the ground truth, (b) identified the weak source and had a lower LBPM metric from the observation. Therefore, it will be preserved for further iterated optimizations. Figure RC2. 2 shows that the LBPM outperforms the RMS in terms of convergence rate, and fitting accuracy (IoU, PWIoU) in a Monte Carlo stochastic test.

In the next revision, we will highlight LBPM's enhanced performance in detecting **relatively weaker** sources and extend the discussion, particularly focusing on potential areas for its limitation and improvement. Additionally, we will refrain

from making somewhat overly optimistic statements.

[Figure]

Figure RC2. 1 Demonstration of LBPM's outperformance over RMS in capturing relatively small

sources in the iteration.

[Figure]

Figure RC2. 2 Preliminary results of plume separation of RMS (left) and RMS (right). IOU and

PWIoU are two separation accuracy indicators.

- *"- the abstract could mention the test on real data which is significant."*

**Response:** Thanks for the suggestion. We will mention it in the next revision.

- *"- l11: MAPE has not been defined yet; you should rather speak about the average relative error in the estimates."*

**Response:** We will add it in the next revision. MAPE represents "mean absolute percentage error".

- *"- l63: how do you derive a "median interval distances of potential methane sources in California" ? I'm curious about it since the result (<200m) is surprisingly small (it is probably a matter of definition for such a term)"*

**Response:** There are 234060 targets given in the VISTA-CA inventory with their locations in longitude and latitude. We project these locations into ESPG: 26941 (a Cartesian 2D axes for California, US, with an accuracy of ~2m). The exact intervals,

Euclidean distance between one source and its nearest neighboring source, are then calculated. Over 90% of the source intervals are less than 200 m (Figure 1); if sources with intervals less than 30 m are merged, ~50% of the spacings are less than 200 m.

However, sources of type "oil and gas well" account for the vast majority (96.5%) of the inventory. If they are excluded, the median and mean interval become 495.9 m and 1247.1 m, respectively. The median is much lower than the mean, revealing the characteristic of spatial clustering distribution of emission sources. We will clarify this in the next revision.

[Figure]

Figure RC2. 3 Distribution of source intervals in VISTA-CA, with (left) and without (right) "Oil and Gas Wells".

- *"- l68-70: unclear; you mean Nassar et al. fix the secondary plume to a specific concentration level corresponding to the secondary source rate given by the emission inventory ? "*

**Response:** Yes, we confirmed this point with Dr. Nassar in an Email. They fix the emission rates of the secondary sources. They also adjust them by $\pm 20\%$ to assess the uncertainty.

Some potential reasons why they cannot estimate multiple sources are listed here. Firstly, OCO-2 observations are too sparse to provide enough constraints to estimate multiple sources (Nassar's opinion). Secondly, their estimation method LS methods are prone to local optima and are thus inferior to heuristic optimization algorithms in finding global optima. Thirdly, Gaussian plume fitting is less robust than the combination of Gaussian plume fitting and IME method, as shown in our experiment (Exp1, Exp2).

- *"=> then you should also discuss other papers using multiple Gaussian plume model inversions to tackle multiple sources with observations similar to satellite images, i.e. something similar to the "Multi-source Gaussian plume" method tested in section 3 (e.g, Krings et al., 2011, which is already cited in the manuscript). I think that the introduction is a bit misleading regarding this: using multiple Gaussian plume model fitting to handle multiple sources is not a novelty. However, by developing an alternative approach with a multi-objective heuristic optimization for the Gaussian plume fitting combined with IME (which is often assumed to behave better than Gaussian plume model inversions when tackling fine resolution images of turbulent plumes) for the quantification, the authors bring new ideas and insights to improve the process of overlapping plumes."*

**Response:** Thank you for pointing out our oversight. We have carefully reviewed the paper by Krings et al. (2011) and confirmed that they indeed employed a multi-point Gaussian plume model. We will organize and merge L68-L83 in the introduction as related work, with a specific emphasis on the contributions of Krings et al. (2011) and Nassar et al., (2017). We sincerely appreciate your summary of our innovation and we will add it to the text in the next revision!

- *"- l96; what does "the improvements of the separation algorithm in missed detection" mean ?"*

**Response:** Thank you for pointing out this oversight. In our latest revision, we have eliminated the pixel connectivity criterion and shifted our focus primarily to the Mean Absolute Percentage Error (MAPE) as the evaluation indicator for quantification, to make the content more concise. This sentence was overlooked during the inspection. We will scrutinize such issues in the next revision.

- *"- l124: what could be the difference between sigma_x and sigma_y when using Pasquill stability classes to set-up such parameters? where could sigma_x (and where does sigma_x) stand in eq (2)? furthermore, the formula for the function sigma_y(x) needs to be given or explained"*

**Response:** Thank you for pointing out this slip of the pen, where $\sigma_x$ is redundant. cross-wind dispersion coefficient ($\sigma_y$) is a function of down-wind distance $x$ and Pasquill stability class (which is a function to **wind speed $u$**, sun illumination, etc), and surface type. $\sigma_y$ is given by a series of functions, which can be directly found in

Table 3. Brigg's parameterization for rural sites

| Stability class | $\sigma_y$ (m) | $\sigma_z$ (m) |
|---|---|---|
| A | $0.22x(1+0.0001x)^{-1/2}$ | $0.20x$ |
| B | $0.16x(1+0.0001x)^{-1/2}$ | $0.12x$ |
| C | $0.11x(1+0.0001x)^{-1/2}$ | $0.08x(1+0.0002x)^{-1/2}$ |
| D | $0.08x(1+0.0001x)^{-1/2}$ | $0.06x(1+0.0015x)^{-1/2}$ |
| E | $0.06x(1+0.0001x)^{-1/2}$ | $0.03x(1+0.0003x)^{-1}$ |
| F | $0.04x(1+0.0001x)^{-1/2}$ | $0.016x(1+0.0003x)^{-1}$ |

Table 4. Brigg's parameterization for urban sites

| Stability class | $\sigma_y$ (m) | $\sigma_z$ (m) |
|---|---|---|
| A–B | $0.32x(1+0.0004x)^{-1/2}$ | $0.24x(1+0.001x)^{1/2}$ |
| C | $0.22x(1+0.0004x)^{-1/2}$ | $0.20x$ |
| D | $0.16x(1+0.0004x)^{-1/2}$ | $0.14x(1+0.0003x)^{-1/2}$ |
| E–F | $0.11x(1+0.0004x)^{-1/2}$ | $0.08x(1+0.0015x)^{-1/2}$ |

Figure RC2. 4 Brigg's definition of dispersion coefficients.

the reference (Griffiths, 1994), as shown in the middle column in Figure RC2. 4. In some research, different definition of dispersion coefficients might be used. For instance, Masters (1998) proposed an alternative formulation, which has been adopted by studies such as Bovensmann et al. (2010) and Nassar et al. (2017); more localized formulations such as China's national standard GB/T 39499-2020 may also be employed. Due to its diversity, we have chosen not to present them explicitly in the text.

- *"- equations 3 and 4: we should have C(x'_n, y'_n) rather than C_n(x', y')"*

**Response:** The transferred coordinates $(x', y')$ can be replaced by (x'_n, y'_n), as they are dependent on the parameters of source n, including emission rates ($Q_n$) and source location ($x_n, y_n$). However, by this convention, the suffix n of $C_n$ shouldn't be omitted, as it represents the concentration caused by a certain source $n$.

- *"- l146 The specific Gaussian plume fitting algorithm used here is questionable. Adjusting both u and Q to fit the observations can be problematic since these 2 parameters impact the plume amplitude in a similar way (there is no source of information to discriminate them in the fitting process). Furthermore, the lack of adjustment of sigma_y in the plume fitting may limit the skill of the approach."*

**Response:** Thank you for inquiring about the fundamental principles of our method. As previously mentioned, the diffusion coefficient determines $\sigma_y$ is a function of distance and atmospheric stability, which is a function mainly of wind speed. Thus, wind speed will indirectly determine the plume shape. As presented by Jongaramrungruang et al. (2019; Figure 9), wind speed determines the magnitude of the "half-mass angle" of the plume. Therefore, when the global optima achieved in the plume fitting, it is possible to decouple the emission rate ($Q$) and the wind speed ($u$).

It is worth mentioning that our LPBM as an optimization objective is to some extent precisely to fit the plume shape, to avoid falling into a false local optimum.

- *"- l170-l175 are very difficult to understand"*

**Response:** We will combine Figure 1 to provide a detailed explanation of the input, output, and implementation process. We will follow the sequence of parameter estimation, Gaussian plume, Gaussian blur, and weighted allocation for the implementation process.

- *"- l281: the question is not whether these sources can be quantified, but rather whether their plumes can perturb the quantification of other sources (?)"*

**Response:** In the early experiments, the plume concentrations of these weak sources are generally similar to the noise level. Therefore, we exclude their analysis to focus on the larger targets, as well as to reduce the difficulty of program development.

- *"- l282: I do not really understand this sentence, and in particular the term "aggressive". It seems to connect to the assumption that the emissions are constant, which makes the quantification problem and the process of the image easier."*

**Response:** Assuming continuous emissions rates in our simulations serves to simplify the problem and increase the probability of plume overlap. We make this "aggressive" simplification as the inventory maker manually removes the overlapping plumes in its quality control phase (Duren et al., 2019). We will briefly state this simplification and its reasons in the next revision.

- *"- there is still a lack of explicit introduction to the fact that the whole study relies on scenes driven by homogeneous winds, except when tackling EMIT*

*observations. It's implicitly guessed from the use of theta rather than theta_n in the equations, and by the quick mention to the wind speed at lines 254 (with the clumsy formula: "unified wind") and 285. There is also a lack of discussion on the fact that the use of homogeneous winds to generate the pseudo observations artificially inflates the skill of Gaussian plume models to support the plume separation: there is a piece of sentence about it at lines 461-462 but it is not very clear and it seems to be associated to very specific observation cases."*

**Response:** Thank you for the suggestion. We employed the homogeneous winds assumption at two key stages:

1. We used homogeneous winds assumption to load LES plumes with the given 10-m winds load from ERA5 to synthesize the pseudo-observations. We will stress this point after Eq. 12.

2. Additionally, we leveraged the assumption of homogeneous winds to constrain the search range, thereby accelerating the convergence. We will emphasize this point after Eq. 4.

We will also extend this point in the discussion.

- *"- it would be useful to show images with different levels of observation noise, in suppl. material if not in main text (the presentation of fig 2 would be misleading if it does not include such a noise: does it ?) to provide an idea of the challenge associated to the plume separation when using noisy images."*

**Response:** Thanks for the suggestion, we will consider the possibility of including these images in the Supplement. In Figure 2, a noise of 1% (18 ppb) to the background concentration has been added. The source emission rates in Figure range from 0.2 to 2 t/h. It's worth noticing that the color bar is truncated, suppressing the noise visually.

- *"Actually, one would expect charts with the APE as a function of the level of noise since this level could be one of the drivers of the relative success of SEP vs. UNSEP. In a more general way, the retrieval uncertainty is a critical topic for the processing of plume images and the manuscript should bring more insights about it."*

**Response:** We tested on single source scenarios in EXP1, where the APE as a function of noise in bar graphs is shown in Figure 1. In EXP2, our focus shifted to quantification comparisons under various overlapping scenarios, where factors directly influencing the overlapping index (OImass) include wind speed, wind direction, distance between two sources, and emission ratios between two sources (shown in Figure 2). The inversion noise in EXP2 was considered as a secondary factor, hence we treated it as fixed. We didn't expect there to be a large difference between SEP & UNSEP in Exp2 compared to Exp1. We will perform a new experiment of different noise levels in Exp2 to finally decide whether it is necessary to add it to the content.

- *"In particular, it should provide the values of the level of noise in a more visible way than at lines 266 and 272, and with some justification for such values."*

**Response:** We will conduct a new experiment to determine the necessity of adding these plume images of different noise levels. The noise levels are mainly from Varon et al. (2018), where the typical retrieval error for GHGSat ranges from 1%-5%.

- *"What is the value of the retrieval uncertainty in EXP-3 ? Is the % applied to the CH4 background + plume signal? If yes, what is the corresponding background value ? If not, why (note that the full EMIT image is noisy in fig 8), and would not the values 1 to 3% be very low ? What are the typical relative uncertainties associated to EMIT observations ? Could the observation noise explain the relative failure of the use of the LBPM metric ?"*

**Response:** In EXP-3, the retrieval uncertainty is set at 1%, expressed as the fraction of the background concentration. The corresponding background is 1.8 ppm & 1 atm (with a concentration ~10.3g/m2), as shown in L230-L231. We focused on the next generation of fine-spatial resolution and hyperspectral satellites, such as GHGSat, capable of obtaining XCH4 with high precision. The relatively limited performance of LBPM may primarily stem from the limited performance of Gaussian plume for such a small scale.

- *"- lines 290-300 poorly fit in section 2.2.3; having separate sections dedicated to the detection / quantification method(s) and to the data made available for the*

*detection / quantification would make the presentation of the study much clearer."*

**Response:** Thank you for the suggestion. In the next revision, we will introduce a new subsection to provide a more detailed implementation of both quantification schemas (UNSEP & SEP) for the observations.

- *"- l294-296: this derivation of the wind driving the plume from the wind at 10 m height does not really make sense for the general process of satellite images. This is likely inherited from studies focusing on specific sites where dedicated local measurement of the wind at 10 m are available. In the general case, the wind should be derived from other source of knowledge (typically meteorological analysis) with a better vertical coverage (but less precision)."*

**Response:** Utilizing 10-m wind speed as the near-surface wind speed for the quantification is a common practice in recent studies within this field, as the direct availability of the 10-m wind components from meteorological reanalysis databases.

For our simulation, the decision to use 10-m wind speed is also driven by the need for simplification, as elaborated in L239-L243. While simulating specific scenarios with real driving wind profiles and other real boundary conditions would undoubtedly offer greater precision, the computational cost can be huge, especially when considering various scenarios. Therefore, we simplify to simulate plumes of various wind speeds and use the 10-m wind speed from ERA5 as an index to load plumes from the WRF-LES database to synthesize the pseudo-observations.

- *"- l297-300: clarify how it is combined with the SEP approach"*

**Response:** In the next revision, we will introduce a new subsection to provide a more detailed implementation of both quantification schemas (UNSEP & SEP) for the observations.

- *"- there is no discussion on the CH4 background mixing ratio fields (from sources outside the images, or from small point sources and diffuse area sources within the images), on how it is dealt with in the derivation of the images or when processing the images. Such a background is set to 0 in EXP 1 to EXP 3. How could it impact the theoretical results from these 3 experiments ? Do we see some residual pattern of background variations in the EMIT "enhancement data" ?*

*could it explain part of the biases seen in section 3.4 ?"*

**Response:** The complexity of the real background, decided by factors such as surface BRDF, instrument characteristics, as well as such "background" of interference concentration, has been stressed previously (Gorroño et al., 2022; Jongaramrungruang et al., 2022). Analysis considering such noise may require a specified instrument and scenario, and it is probably beyond the scope of this article. Therefore, currently, in our experiment, we only considered additive random noise.

Although such a "background" wasn't presented in the Exp1-3, it's important to note that the unexpected enhancements in the "background" might slightly inflate the plume concentration. However, it's still premature to conclude whether it will finally inflate the IME and Q, as the "background" may also influence pixel detection, introducing non-linearity and complicating this issue.

The monotonous surface type (desert) in Figure 8 and the relatively large pixel size of EMIT (~60 m) might help alleviate the interferences from unexpected surface features. However, as shown in Figure RC2. 5, there is a large portion of pixels identified as plume pixels by the UNSEP and rejected by the SEP. It is difficult to tell them the continuity of an up-wind plume or a new plume. This explanation could partially explain why the quantification results of the whole cluster by UNSEP are slightly higher than the result by SEP.

- *"- isn't line 322 at odd with equation 14 ?"*

**Response:** Thank you for noticing this error in the text. Eq.14 is correct and the text in L322 should be "the ratio of plume mass integration from interference sources to that from the primary source". We will correct it in the next revision.

- *"- line 328: I do not understand why "the quantification of methane source is considered as solving a regression problem". Is not the target of such a quantification the most precise estimate for a given source at a given time ?"*

**Response:** Thank you for the suggestion. It should be solving a parameter estimation problem.

- *"- l367: so far, the metric for interference should be the OImass, not "the interference" "*

**Response:** Thank you for pointing out this glitch. We will correct it in the next revision.

- *"- l393: Fig 6 rather gives the feeling that UNSEP tends to overestimate the sources ? you meant SEP ? what do the person's R and p values correspond to in this line ?"*

**Response:** We apologize for this slip of the pen. It should be SEP in L393. Pearson's *R* is the correlation coefficient between the ground truths and the predictions of source emission rates. we will correct it in the next revision.

- *"- I do not understand the correction of the SEP estimations in section 3.4 (l393-395, legend of fig 6): what is the rationale, what is done in practice ? Because of this correction, it is difficult to check whether SEP behaves better than UNSEP in section 3.4."*

**Response:** As shown by Figure 6 (b), there was a systematic underestimation and we found this underestimation is linear (R=0.93, p<0.01). After a linear regression, there is $y = 0.51x - 1.31$, where $y$ represents the quantification result of the SEP; and $x$ represents the ground truth emission rates. So, the corrected quantification result can be simply expressed as $1.96y + 2.57$. The distribution of SEP is more concentrated, especially for source targets with low emission rates.

- *"- l413 vs l422: does the detection and "connectivity verification" underlying the application of the IME method really encompass the full extent of the whole set of plumes ?*

**Response:** In non-overlapping scenarios, the main structure is typically covered. Connectivity verification, a straightforward attribution strategy, forms the foundation of the applying IME method to scenarios with plume overlapping. However, there can be plume confusion for unseparated overlapping plumes. Moreover, this strategy may omit significant portions, as illustrated in Figure RC2. 5, where a substantial number of pixels identified as plume pixels by UNSEP are excluded by SEP.

[Figure]

Figure RC2. 5 Whole plume pixels (yellow, identified by SEP) and separated & attributed plumes (Viridis coloring; by UNSEP) in the EMIT observation.

- *l413 goes too fast so it is difficult to understand what it corresponds to."*

**Response:** In L413, in comparison to our separated quantification approach, we also employed the traditional IME method by Varon et al. (2018) to quantify the entire plume cluster, denoted as "whole". We will make it clearer for this point by elaborating on a more detailed implementation of two quantification schemas for the observations in a new subsection in the next revision.

- *"- l447-448: I do not understand the point which is made here, while I believe that the comparison between the Gaussian plume model inversion and IME for the emission quantification as a function of the spatial resolution and scale of the source quantification problem is an important topic"*

**Response:** We agree with the opinion and we tried to provide this comparison in the first experiment in 3.1. Our experiments demonstrate that for the quantification of small-scale plumes (within the $6 \times 6$ km$^2$ simulation domain), the IME method generally outperforms the Gaussian plume fitting (GPF). Limited by the computation and the scale of the simulation, a discussion for the comparison on a large scale seems beyond the scope of our current experiments.

However, it's worth noting, as shown by Varon et al. (2018), that the performance of Gaussian plume fitting increases with pixel size on a larger scale. L447-L448 tried to explain why we have not yet found the turning point of performance increase of the GPF. One explanation is that our simulation domain is too small, and the number of pixel samples begins to decrease as we increase the pixel size, which counteracts the benefit of the averaging of eddies by large pixels for performance improvement of the GPF. We will make this point clearer in the next revision.

- *"- l450 I do not understand "statistically correct", and the link made at lines 450-*

*451 between the precision of the emission quantification using Gaussian plume*

*fitting and the convolution kernel"*

**Response:** The LES simulated plume demonstrates Gaussian plume morphology after temporal averaging (Figure RC2. 6), and the LES simulation shows that a transient plume may deviate from the Gaussian behavior (Figure RC2. 7). In our initial experiments, we found that the fitted Gaussian plume could be "spiky" near the source, resulting in missed capturing of some near-source-parts of the transient plume in some cases. To address this, we employ Gaussian blurring (i.e., convolving the image with a Gaussian kernel.), a subtle computer vision technique, to better capture these transient plumes when using Gaussian plume results as weights for the separation. Figure RC2. 8 illustrates how the Gaussian-blurred plume demonstrates more "diffusive" than the original Gaussian plume, particularly near the emission source. We will refine the text in the next revision.

[Figure]

Figure RC2. 6 Time-averaged plumes by WRF-LES. From left to right: 500, 800, and 1100 m of

inversion height. From top to bottom: 1, 3, 5, 7, 9 m/s of wind speed.

[Figure]

Figure RC2. 7 Snapshots of transient plumes by WRF-LES.

[Figure]

Figure RC2. 8 Snapshot of LES-simulated transient plume (left), fitted Gaussian plume (middle),

blurred Gaussian plume (right).

- *"- I do not understand line 471"*

**Response:** As we mentioned in previous response, as well as shown in Figure 1, the separated plumes don't rely on a specific plume detection and quantification method. So, a new plume pixel detection method and new quantification method can be introduced to quantify the separated plumes. We will make this point clearer in the next revision.

**Reference**

- Bovensmann, H., Buchwitz, M., Burrows, J. P., Reuter, M., Krings, T., Gerilowski, K., Schneising, O., Heymann, J., Tretner, A., & Erzinger, J. (2010). A remote sensing technique for global monitoring of power plant CO2 emissions from space and related applications. Atmospheric Measurement Techniques, 3(4), 781–811. https://doi.org/10.5194/amt-3-781-2010
- Griffiths, R. F. (1994). Errors in the use of the Briggs parameterization for atmospheric dispersion coefficients. Atmospheric Environment, 28(17), 2861–2865. https://doi.org/10.1016/1352-2310(94)90086-8
- Jongaramrungruang, S., Frankenberg, C., Matheou, G., Thorpe, A. K., Thompson,

D. R., Kuai, L., & Duren, R. M. (2019). Towards accurate methane point-source quantification from high-resolution 2-D plume imagery. Atmospheric Measurement Techniques, 12(12), 6667–6681. https://doi.org/10.5194/amt-12-6667-2019

- Jongaramrungruang, S., Matheou, G., Thorpe, A. K., Zeng, Z.-C., & Frankenberg, C. (2021). Remote sensing of methane plumes: Instrument tradeoff analysis for detecting and quantifying local sources at global scale. Atmospheric Measurement Techniques, 14(12), 7999–8017. https://doi.org/10.5194/amt-14-7999-2021

- Masters, G. M. (1998). In Introduction to environmental engineering and science (2nd edn, p. 413). Prentice-Hall, Inc.

- Nassar, R., Hill, T. G., McLinden, C. A., Wunch, D., Jones, D. B. A., & Crisp, D. (2017). Quantifying CO2 Emissions From Individual Power Plants From Space. Geophysical Research Letters, 44(19), Article 19. https://doi.org/10.1002/2017GL074702

- Nottrott, A., Kleissl, J., & Keeling, R. (2014). Modeling passive scalar dispersion in the atmospheric boundary layer with WRF large-eddy simulation. Atmospheric Environment, 82, 172–182. https://doi.org/10.1016/j.atmosenv.2013.10.026

- Varon, D. J., Jacob, D. J., McKeever, J., Jervis, D., Durak, B. O. A., Xia, Y., & Huang, Y. (2018). Quantifying methane point sources from fine-scale satellite observations of atmospheric methane plumes. Atmospheric Measurement Techniques, 11(10), 5673–5686. https://doi.org/10.5194/amt-11-5673-2018

---

## Author Response (AR2)

**Author's response**

Dear reviewer and associate editor,

We appreciate the positive attitudes and constructive comments provided by the reviewer and associate editor regarding our manuscript - *Separating and Quantifying Facility-Level Methane Emissions with Overlapping Plumes for Spaceborne Methane Monitoring*.

We have primarily clarified the methods description, definitions related to wind, and also addressed other concerns. Please see the point-by-point response as follows. We hope our refined manuscript will meet the high-standard requirement of AMT.

**Response to Reviewer**

- *"I thank the authors of "Separating and Quantifying Facility-Level Methane Emissions with Overlapping Plumes for Spaceborne Methane Monitoring" for the substantial update of this manuscript following the reviews and for their detailed answers to my comments, even when they have removed the corresponding sections in the manuscript. I also thank them for their transparency with regard to the bias in the comparison between the Gaussian plume model fitting and the IME quantification methods due to the limitation of the image size in their experiments."*

**Response:** Thanks for the approval. Your suggestions has greatly contributed to the improvement of our manuscript and we are delighted to mention it in the acknowledgement.

- *"Many of the updates are satisfying. However, I still push for a revision of the manuscript, because even though they partially tackled them, going in the right direction, the authors did not fully address some of the comments I have raised in the previous review. Here, I do not copy paste these comments from the 1st review, but I refer to them in the following:"*

**Response:** Thanks for reviewing our manuscript with patience and providing thorough and helpful feedback once again. I apologize for the misunderstandings in the last revision and hope the refinement this time can address all the comments adequately.

- *"I think that the abstract, introduction, section 2.1 and parts of the conclusion still lack a very clear and explicit overview of the split of the image processing into separation or "attribution" - detection (including the extraction of the enhancements above the background corresponding to the plumes) - quantification => the three steps that the authors promise to discuss in their answers to my comments, but which do not really appear as such in the new manuscript"*

**Response:** Thank you for suggesting we emphasize the description and discussion of the methods by these three stages. We believe this approach will significantly improve the clarity of our manuscript, and we intend to give it more elaboration in the content this time. We also realize that this could serve as the first clear distinction in this field, potentially benefiting future work.

- *"the 4 different combinations of methods for these 3 steps that will be tested:the method developed here, i.e., the sequence of Gaussian plume model fitting for separation, student's t-test for detection and then the IME for quantification vs. the sequence of student's t-test for the detection, the "pixel connectivity analysis" for the "attribution", and then the IME for the quantification vs. the single Gaussian plume model fitting, ignoring the problem of separation, and solving for the "attribution", detection and quantification all together (still extracting the background before this process ?) vs. multiple Gaussian plume model fitting, solving for the 3-steps all together, taking the separation problem as a problem of solving for several sources at once. => a table somewhere in section 2 may help clarify things"*

**Response:** We sincerely appreciate the direct suggestion. Your suggestion to add such a table helped clarify our manuscript's elaboration. We have included such a table (shown as table 1) in the manuscript. We have divided the methods into three stages, including (1) separation, which isolates overlapping plumes into individual images,

each containing a single plume; (2) detection, which distinguishes the plume pixels from the background pixels; and (3) quantification, which calculates the emission rates of the point sources based on the identified pixels.

**Table 1.** Comparison of methods evaluated in this work.

| Method | Separation | Detection | Quantification |
|---|---|---|---|
| Single-source Gaussian plume | / | Single-source Gaussian plume fitting | |
| Multi-source Gaussian plume | | Multi-source Gaussian plume fitting | |
| UNSEP | / | Student's t-test & Connectivity filtering | IME method |
| SEP (ours) | Gaussian plume weighting separation | Student's t-test & Connectivity filtering | IME method |

Table RC 1 Method combinations for quantification in the manuscript.

- *"the abstract should mainly clarify the separation between separation with Gaussian plume model and quantification with IME (line 4-5 misses something like "respectively" for this) and the alternative use of Gaussian plume model fitting for both (info missing at line 11). The other sections should bring this overall picture quite early before entering into technical details. In the introduction, lines 50-69, 73-79 and then 80-81/88/89 mix everything. The introduction of section 2.1 is focused on the new method and none of the following subsections will provide a clear overview highlighting in a distinct way the "3 steps" and the alternative methods: from 2.1.1, the text jumps into technical details."*

**Response:** Thanks for the suggestion. We refine the text for clear distinction in the abstract. We also improve the introduction accordingly. More importantly, we adopt the three-stage description and improve Section 2.1 with much clarity. We name our separation method as Gaussian plume weighting separation and make a clear distinction between our method and the other three method. We also clearly state the uniqueness of the detection method for IME, and the detection method is not considered a variant in this work.

- *"I think that the presentation of the "pixel connectivity analysis" could be improved: clarifying the fact that it attempts at "separating" plumes which do not overlap, but that it would merge all overlapping plumes into a single one (unless it*

*includes some level of separation of overlapping plumes ?) ? I think that calling it*

*a "pixel attribution" method is a bit misleading from that point of view."*

**Response:** Thanks for the advice. We rename this approach as connectivity filtering, which follows the t-test as the detection approach (shown in Table 1).

- *"- regarding the implicit optimization of sigma_y when optimizing the wind speed u in the frame of the Gaussian plume model fitting => the text is not really clear about this, sigma_y could have been fixed offline with an initial value given to u; so I think that the text should state it, and maybe the notation sigma_y(x) could be changed into sigma_y(x,u) to highlight it better ?"*

**Response:** Thanks for the advice. It's a good idea to highlight that $\sigma_y$ is partially decided by wind speed $u$. However, this expression may be misleading that $\sigma_y(x,u)$ is a continuous function of $u$. In many definations, the $\sigma_y(x)$ functions are given in lists based on different terrain and stability (which is decided discretely by u and sunlight, i.e., decided solely by $u$ in middle sunlight). To be rigorous, it should be written as $\sigma_y(x; u, \text{underlying condition}, \text{sunlight})$. For conciseness, we will omit the parameters in the mathematical formulas and clarify in the text that "$\sigma_y$ represents the diffusion coefficient across-wind, is a function of downwind distance $x$ and is decided by wind speed, underlying condition and sunlight (Briggs, 1973)."

- *"- observation noise: you should provide typical values using the same units (g/m2) as when plotting the plumes and compare them to the typical amplitude of the observed plumes.*

**Response:** It's a reasonable suggestion. However, the noise value in the concentration column (g/m²) may vary with ground pressure and humidity. Typically, it is expressed as a percentage of the dry air column. In this domain, concentrations are usually expressed in parts per billion (ppb), so the unit is often given as ppb or a percentage. To make the content clearer, we've mentioned the typical value of noise level in g/m2 (Please see L217).

- *I think that section 3.2 should refer to the new supplementary material and summarize the conclusions from this supplementary material."*

**Response:** Thanks for the advice. We will mention it briefly in Section 3.2. The reason

why we skip elaborating is that (1) those supplementary results are very similar to those in the content; (2) we focused on the environmental influence on the overlapping in exp2, and the observation factors don't directly decide the overlapping in our experiment. Please see L354-L355.

- *"- regarding the wind: the text is not clear; once having introduced equation 1 (l. 110-111), the authors directly speak about "the" 2D wind vector (implicitly: about the 2D effective wind) as if there was a clear definition for such a 2D wind, and later they have values for the true 2D wind (e.g. at lines 149-151, and then after equation 7, where they fit it with a log function of U10). However, the derivation of the effective 2D wind driving the 2D plumes seen from space can be a complicated topic. How do the authors get it when tackling the 3D LES simulations (it seems that they implicitly assume it to be the geostrophic wind, because of the similarity between the ranges at lines 206 and 250, which would raise questions) ? Could the optimal wind derived from the Gaussian plume model fitting differ from what the author assume to be the effective wind, because of a wrong derivation a priori of this effective wind ? In this case, may the use of the log function of U10 in the IME instead of the wind retrieved from the Gaussian plume fitting not be the best option ?"*

**Response:** Thanks for the advice. Our experiment involves four types of winds: (1) geostrophic wind, driving the LES simulation; (2) 10 m wind, loaded from a meteorological database to calculate effective wind speed for IME; (3) effective wind for IME; (4) effective wind for Gaussian plume. Similar to Varon et al. (2019), we derived the IME effective wind using the 10 m wind speed.

The Gaussian plume effective wind speed can be a relatively independent topic. In previous studies, such as Nassar et al. (2017), effective wind speeds are typically considered as the wind speed at the emission height. The effective wind speed for the Gaussian plume is coupled with the dispersion coefficients, defining the plume shape. Often, the dispersion coefficients are highly empirical, making the optimal wind speed differ from the actual horizontal wind speed at the emission height. In our application, our main objective is to fit the plume shape using the Gaussian plume model. Therefore,

we use the optimal wind speed to mitigate the impact of inaccurate dispersion coefficients and achieve good agreement between the modeled and observed plume. Additionally, the relationship between effective wind speeds for the Gaussian plume and IME requires further discussion, as the definition of IME can also be empirical.

We also want to clarify the basis for matching 10 m and geostrophic wind speeds in exp3, where plumes are loaded according to ERA5 10 m wind speeds. As shown in Figure RC 1, our LES simulations established a boundary layer, and the relationship between geostrophic wind and 10 m wind can be expressed as $u_{10} = 0.86u_g + 0.26$ (Figure RC 2). Thus, we can match $u_g$ using $u_{10}$ from the meteorological database, given by $u_g = 1.16u_{10} - 0.30$. More details are elaborated in the supplement.

[Figure]

Figure RC 1 Distribution of horizontally averaged wind speed in the LES simulation.

[Figure]

Figure RC 2 Relationship between the geostrophic wind ($u_g$) and 10 m wind ($u_{10}$) in the LES simulation.

- *"There was some misunderstanding regarding the background: in my comment, I was not speaking about the variations due to errors in the CH4 concentration retrievals. I was speaking about the impact of the CH4 sources located outside the satellite image. The frequent vicinity of sources within the images implies the frequent vicinity of other clusters of sources outside the images whose plumes all together may raise larger problems of overlapping than the sources within the image. There could also be areas sources close to the targeted sources, whose atmospheric signal overlaps the global background. If such a problem has been ignored, it should be explained in the method section. The manuscript should clarify whether the detection-quantification steps assume that the background field is uniform to derive the CH4 "enhancements" (the term is used but not really defined; it corresponds to the enhancements above the background). Lines 144-146 (and equation 5) completely ignore the background field (which adds to the general problem of the lack of overview on the "3 steps" to process the images that is discussed above)."*

**Response:** Thanks for the detailed clarification on the suggestion. We introduce the term "CH4 enhancements" directly from other research in this field. In this work, we consider the background uniform, and the plumes and noise solely contribute to the

enhancements. The influences of the methane background are not considered as this topic is rather complicated and beyond the scope of this work. The plumes of near sources originating from outside the domain are considered well-mixed and thus can be treated as background, as supported by the results of Exp2. These sources do not contribute to the enhancements. We will clarify this point in the content.

- *"regression problem vs. parameter estimation problem in section 2.4.2: replacing the former by the latter does not solve for the issue here since the text still states in the next sentence "So, the R2 coefficient of determination is introduced to indicate the accuracy of overall estimation results", and at line 308 "regression results"*

**Response:** Thank you for pointing this out. We do not intend to debate the relationship between parameter estimation and regression problems, though these concepts are often mixed and not distinguished, with regression being performed through parameter estimation. To maintain coherence with the literature, we will refine the text accordingly.

- *"I still do not understand the "correction" of the SEP results in section 3.3: I understand that debiasing the results improve their accuracy, but what authorizes the authors to apply such a correction which is completely based on the knowledge of the true emissions ? What is the applicability of such a correction if considering experiments with real images ?"*

**Response:** Thank you for highlighting this issue. The introduction of Gaussian plume weighting can dilute the plume image and decrese the IME value. Therefore, when applying the IME method, $u_{eff}$ needs recalibration with different plume pixel detection method. Previously, we used a "post-calibration" approach.

We are now pleased that we have solved this problem by introducing an additional calibration for $u_{eff}$ with Gaussian plume separation practice, and the result is $u_{eff} = 0.94 + 0.64 \log(u_{10})$. In comparison, the result without separation is $u_{eff} = 0.62 + 0.55 \log(u_{10})$. We reprocessed the original results of exp3 and the new results of exp3 are shown in Figure RC 3 (Figure 6 in the manuscript). The present R2 and MAPE indicators are also comparable to previous results with "post-calibration". Necessary modifications are also made for the corresponding section.

[Figure]

Figure RC 3 Comparison between quantification results of (a) unseparated quantification and (b) separated quantification in Exp3.

- *"- equations: C_n must be defined mathematically to be used in the right hand side of eq3; I still have the feeling that eq3 works with C instead of Cn in this right hand side, and, actually, line 130-131 is not consistent with the current eq3; l. 174: just say Cp is the modeling of plume p, but anyway, this definition has already be given around eq3 (don't redefine it several times; actually, C is redefined plenty of times throughout the manuscript); "*

**Response:** In our previous manuscript, $C$ represents the concentration modeled by single Gaussian plume; $C_n$ also represents concentration modeled by single Gaussian plume but with specified parameters for source $n$; $C_N$ represents the summation of single Gaussian plumes, i.e., the multiple Gaussian plumes. It also seems that there are naming conflicts for the modeled concentration and observed concentration.

To eliminate the ambiguity, we will follow the following definations: (1) $C_{SG}(x, y; u, Q)$ or $C_{SG}$ represents general single single Gaussian plume; (2) $C_{SG,i}(x_i', y_i'; u_i, Q_i)$ or $C_{SG,i}$ represents a specified single single Gaussian plume of source $i$; (3) $C_{MG}$ represents the multiple Gaussian plume model; (4) $\Delta\Omega$ represents the observed column concentration.

- *"equation 8: should not you write i'_n j'_n ? should not eq8 look as similar as eq 3 and eq4 as possible (for the sake of clarity) since it's a similar process ? line 334: isn't it i' and j' rather than i and j ?"*

**Response:** Thank you for the suggestion, we will improve it accordingly.

- *"Please rewrite sentences such as: "Although there have been abundant spaceborne methane observations, these observations suffer from the demerit of the lack of priors" (l 197), "The LES run by WRF is thus a preferred option for spaceborne GHG monitoring ." (l 201-202)"*

**Response:** Thanks for the advice. Please see the refined text at L199-L202.

---

## Author Response (AR3)

**Author's response**

Dear reviewer and associate editor,

    We appreciate the positive feedback by the reviewer and associate editor regarding our manuscript - *Separating and Quantifying Facility-Level Methane Emissions with Overlapping Plumes for Spaceborne Methane Monitoring*.

    Please see the point-by-point response as follows.

**Response to Reviewer**

- *"I thank the authors for their acknowledgements and for their new corrections, which now address the concerns I raised in the previous reviews.*

  **Response:** Thanks for the approval and thanks again for the precious suggestions.

- *I have only one remaining point: the one regarding the effective wind for the application of the IME method (lines 185-188). The answer of the authors is a bit unclear regarding this. They start to agree that there is a difference between "the geostrophic wind driving the LES simulation" and "the effective wind for IME". However, then, they simply state " Similar to Varon et al. (2019), we derived the IME effective wind using the 10 m wind speed. " without, first, explaining how they derive the reference value for the effective wind (the fitted one) allowing to establish the relationship "Ueff = 0.55logU10 + 0.62 for UNSEP, and Ueff = 0.64logU10 + 0.94 for SEP". Varon et al., 2018 use the knowledge of the sources in the simulations to derive Ueff as Ueff =QL/IME in a set of reference cases. Do the authors conduct such a computation ?*

  **Response:** We derive the effective winds using two simple Monte Carlo simulations. Similar to Varon et al. (2019), we generate large sets of plumes with varying emission rate $q$, 10 m wind speed $U10$. We then calculated the plume characteristic length $L$ and the *IME* using the detected plume pixels by two methods. Given the definition *Ueff =QL/IME,* we fit the parameters a and *b* for $U_{eff} = a \times \log U_{10} + $

*b* to the generated datasets for each method, as shown in Figure 1. We will add a brief description in the next revision.

[Figure]

Figure 1 Ueff and 10 m wind speed.

- *This point about the definition of reference Ueff values and of their fit with a linear function of logU10 gets even more critical than before since the discussion about section 3.3 has pushed for a "recalibration" of Ueff for SEP, which led to a strong improvement of the results in this section, but which is a bit unclear to me. In practice, what does this recalibration correspond to ?*

  **Response:** The reason why the parameters for effective winds differ is that factors such as the plume pixel detection method may underestimate the IME by excluding plume pixels with low concentration. The relation of emission rates *Q* with respect to *Ueff*, *IME* and *L*. If we focus only on fitting *Ueff,* it needs to be fitted for different pixel-detecting methods respectively. As the pixel-detecting process such as the Gaussian plume weighting dilutes the plume image and decreases *IME/L* compared to the thresholding method; as result, the fitted slope and bias of *Ueff* for SEP are higher than those for UNSEP.

  To name this approach "recalibration" in the previous response may be linguistically misleading. Our focus is to fit the parameters of the *Ueff* function separately for each detection method.

- *I also add few minor suggestions for the abstract:- line 8: complement "of the IME method" by something like " when applied without such a separation "*

  **Response:** Thanks for the suggestion. We will add it in the next revision.

- *- line 11: complement " the proposed method " by something like " the application of the proposed separation method together with the IME quantification approach "*

  **Response:** Thanks for the suggestion. We will add it in the next revision.

- *- line 14: try to modify/complement " the more precise single-point source quantifying algorithms, the IME method, " to say (if you want) that IME is currently, probably, one of the most precise single-point source quantification algorithm when tackling high resolution images with turbulent plumes, but to avoid making this statement too general (when considering images of plumes at mesoscale, the IME approach does not appear to be the most precise approach)''*

  **Response:** Thanks for the suggestion. We will modify the text accordingly in the next revision.